# Modeling the language cortex with form-independent and enriched representations of sentence meaning reveals remarkable semantic abstractness

## Abstract

The human language system represents both linguistic forms and meanings, but the abstractness of the meaning representations remains debated. Here, we searched for abstract representations of meaning in the language cortex by modeling neural responses to sentences using representations from vision and language models. When we generate images corresponding to sentences and extract vision model embeddings, we find that aggregating across multiple generated images yields increasingly accurate predictions of language cortex responses—sometimes rivaling large language models. Similarly, averaging embeddings across multiple paraphrases of a sentence improves prediction accuracy compared to any single paraphrase. Enriching paraphrases with contextual details that may be implicit (e.g., augmenting "I had a pancake" to include details like "maple syrup") further increases prediction accuracy, even surpassing predictions based on the embedding of the original sentence, suggesting that the language system maintains richer and broader semantic representations than language models. Together, these results, enabled by a novel modeling framework that leverages both visual and enriched paraphrastic representations, characterize the degree of abstractness in meaning representations within the language cortex.

## 1 Introduction

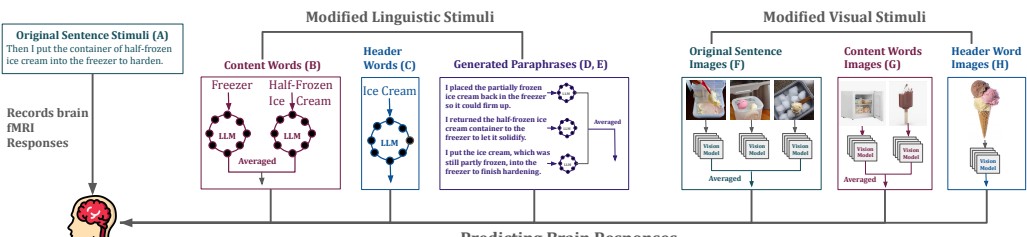

Figure 1: We probe whether neural activity in the language cortex can be explained by different representations of the original linguistic input. Starting from each original sentence, we derive and analyze eight alternative representations - (A) original sentence, (B) content words capturing the main semantic elements, (C) a header phrase summarizing a group of related sentences, (D) paraphrases of the original sentence, (E) paraphrase of the original sentence enriched with commonsense context, (F) images generated from the original sentence, (G) images generated from the content words, and (H) images generated from the header phrase. Language-based variants (A–E) are embedded with large language models, while visual variants (F–H) are embedded using vision models.

In recent years, Large Language Models (LLMs) have shown remarkable potential in modeling neural activity in the human language cortex. A central approach for studying these brain–model correspondences has been encoding models, which predict neural responses from features of linguis-

tic stimuli. Early studies relied on corpus derived, automated and hand-constructed feature spaces including word embeddings, phonemes, syntactic structure, or narrative properties distinctions to construct encoding models of the language cortex Mitchell et al. (2008); Wehbe et al. (2014a); Huth et al. (2016); De Heer et al. (2017). These works demonstrated that word-level representations could account for meaningful variance in brain responses, but often ignored the broader sentence context in which words are embedded. Subsequent efforts showed that incorporating context, capturing relationships between words across time leads to improved modeling of language cortex activity Wehbe et al. (2014b); Jain & Huth (2018). Building on these findings, recent studies have turned to large artificial neural network–based language models, demonstrating that modern LLMs optimized for next-word prediction (e.g., GPT-style models) or trained with contextual objectives (e.g., masked language models such as BERT) provide SOTA predictions of neural responses and reveal striking convergence between artificial and biological language systems Toneva & Wehbe (2019); Schwartz et al. (2019); Schrimpf et al. (2020); Goldstein et al. (2022); Toneva et al. (2022); Hosseini et al. (2024); Tuckute et al. (2024a).

Next, researchers have sought to understand which components of language drive the similarity between model representations and brain responses. A recent study demonstrated that semantic meaning, rather than surface-level lexical or syntactic features, is the dominant factor underlying brain–model alignment Kauf et al. (2024). Research all along has also consistently upheld the importance of preserving the sentence form for activating the language cortex: original sentences reliably yield higher responses than word-level manipulations (e.g., word lists or paraphrases), highlighting the role of meaningful compositional structure. However, to our knowledge, there has been little systematic investigation of whether alternative representations (alternating either in form or even modality) that preserve semantic content can similarly predict language cortex activity.

A broader perspective comes from recent work on representational convergence across neural networks. (Huh et al., 2024) proposes that models trained with different architectures, objectives, and even modalities often converge onto shared high-level semantic structure. Related findings in visual neuroscience echo this theme: activity in higher-order visual cortex can be predicted surprisingly well from linguistic scene descriptions, with captions in some cases outperforming detailed visual features (Saha et al., 2024; Wang et al., 2022; Doerig et al., 2022; Tang et al., 2023; Conwell et al., 2023). These results suggest that certain brain regions and certain artificial models encode meaning at a level abstracted away from modality-specific details. This naturally raises the question of whether analogous forms of representational flexibility might exist within the language cortex itself. Yet such generalization is not guaranteed: similarity between vision and language models, or between linguistic descriptions and visual cortex responses, does not imply that the language-selective network should share that same structure. Whether the language cortex represents meaning in a format that can be approximated by diverse, nonlinguistic sources remains an open empirical question.

**In this paper, we take up this question directly: we examine whether the information encoded in the human language cortex can be modeled from diverse representational sources that differ substantially from the original linguistic stimulus in both form and modality.** We also explore the role of commonsense knowledge, information that is implicit to humans but not explicitly present in the sentence in shaping cortical representations. Our key contributions are as follows:

1. We propose a novel encoding approach for modeling the language cortex in which, instead of using the original linguistic inputs associated with the recorded brain responses, we use inputs that vary in both form and modality but preserve the underlying meaning.

2. We demonstrate that embeddings from vision foundation models, applied to visual depictions of sentences, possess non-trivial predictive power for modeling language cortex activity during sentence comprehension. Importantly, this predictivity increases when incorporating multiple diverse images per sentence, demonstrating that representational averaging provides a closer approximation of the format of linguistic meaning in the brain.

3. Next, we demonstrate that foundational large language model embeddings of paraphrased sentences also predict language cortex activity, with accuracy improving as more paraphrases are used. This mirrors the visual domain findings: greater representational diversity enhances prediction accuracy through averaging, indicating that the language cortex can be modeled even when stimulus form differs substantially from the original input.

4. Finally, we show that enriching sentences with commonsense context-information evident to humans but not explicitly present in the original text produces a substantial boost in predictivity. This finding underscores the critical role of implicit background knowledge in shaping cortical representations and suggests that effective brain-model alignment requires integrating structured, contextually relevant knowledge beyond surface-level representations.

## 2 TRAINING AND DATASETS

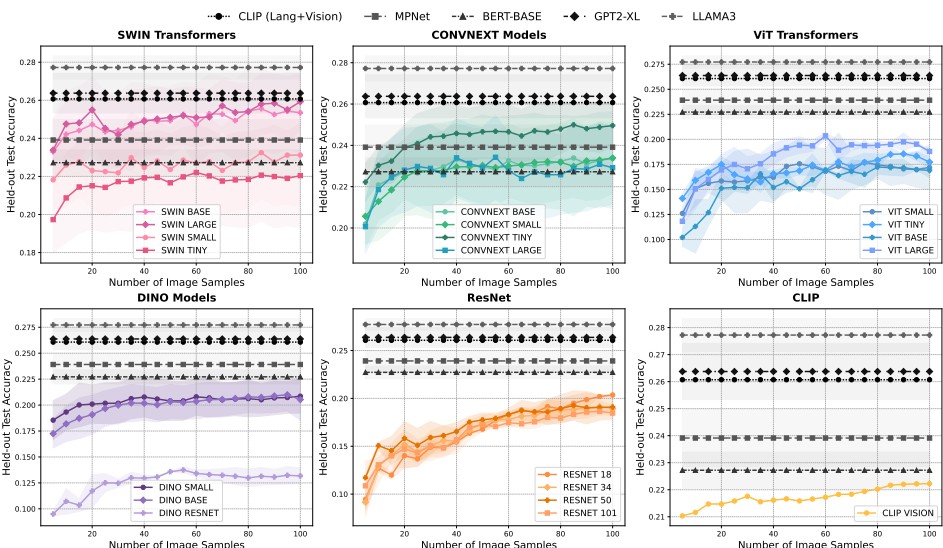

Figure 2: Comparison of performance in predicting language cortex activity between LLM embeddings of the original linguistic stimuli from the Pereira (2018) dataset (CLIP, MPNet, BERT-BASE, GPT2-XL and LLAMA3 presented in grey horizontal lines) and vision model embeddings of their corresponding visual counterparts (remaining models). Performance of the visual models increases with the number of images, sometimes surpassing some of the language models.

We analyzed voxel responses from the 'core' language network, comprising the Inferior Frontal Gyrus, Inferior Frontal Gyrus – Orbital part, Middle Frontal Gyrus, Anterior Temporal cortex, and Posterior Temporal cortex across three datasets where participants read and processed sentences during fMRI scanning: Pereira (2018) Pereira et al. (2018), Tuckute (2024) Tuckute et al. (2024b) and Caption Scene Dataset CSD (2025) Li et al. (2025). A detailed description of all datasets is provided in Appendix Section A.1.

We first investigate whether meaning in the language cortex is captured exclusively by language model embeddings of the original sentence, or whether it can also be represented by alternative sentences that convey the same information. We also test whether representations from other modalities, such as vision, can capture this meaning (Figure 1). Finally, we assess whether predictive accuracy based on the original sentence can be further improved by enriching it with commonsense contextual information.

To evaluate predictive accuracy, we fit a ridge regression model $\boldsymbol{Y} \approx \boldsymbol{E}\boldsymbol{W}$ by minimizing $\|\boldsymbol{Y} - \boldsymbol{E}\boldsymbol{W}\|_F^2 + \lambda\|\boldsymbol{W}\|_F^2$, where $\boldsymbol{Y}$ denotes the voxel responses, $\boldsymbol{E}$ the embeddings from either vision or language models, and $\boldsymbol{W}$ the regression weights (examples in Appendix Figures 10, 12, 11, 14 and 15). The parameter $\lambda > 0$ is the regularization hyperparameter, chosen via cross-validation to prevent overfitting. A more detailed description on training can be found in Appendix Section A.2. We use the following featurizations (different variations of $E$):

A. **Original sentences:** We use the original sentence $\{S_i\}_{i=1}^N$ presented to the subjects, obtain the penultimate layer embeddings from an LLM as $\boldsymbol{E} = \{\boldsymbol{PL}(S_i)\}_{i=1}^N$ where $\boldsymbol{PL}(\cdot)$ denotes the penultimate layer embedding.

B. **Content words:** We extract the most concrete and semantically salient terms $\{c_j\}_{j=1}^{C^S}$ from each sentence: open-class parts of speech such as nouns and main verbs carrying the core meaning of the sentence (Appendix Figure 12). They are identified using a SciPy-based syntactic parser. For example, from "The boy is eating pancakes", we extract "boy" and "pancakes". Each content word is embedded separately using the penultimate layer of the LLM, then averaged to create a sentence-level representation: $\boldsymbol{E} = \frac{1}{C_S}\Sigma_{c_j}\boldsymbol{PL}(c_j)$.

C. **Header words:** For the Pereira (2018) dataset Pereira et al. (2018), sentences are grouped into paragraphs sharing a common topic or header (Appendix Figure 11). We use the paragraph header $\boldsymbol{H}$ as a high-level semantic summary. The header embedding $\boldsymbol{E} = \boldsymbol{PL}(\boldsymbol{H})$ is used to predict the averaged brain responses $\boldsymbol{Y} = \frac{1}{K}\Sigma_{j=1}^{K}\boldsymbol{Y}_{S_j}$ across all sentences in the paragraph, where $\boldsymbol{Y}_{S_j}$ is the brain response for sentence $S_j$ and $K$ is the number of sentences in the paragraph. Note that in the Pereira (2018) dataset's third experiment, several paragraphs described the same topic, we therefore merged all such paragraphs and used the shared topic as the paragraph header.

D. **Standard paraphrases:** We generate $\boldsymbol{R} = 70$ paraphrases (Appendix Table 1) for each sentence using Gemini Comanici et al. (2025) (alternative phrasings that preserve the original semantic meaning while varying in surface form) (Appendix Tables 14) and 15. To analyze how the number of paraphrases affects prediction accuracy, we systematically sample subsets in increments of 5 (i.e., $r \in \{5, 10, 15, ., 70\}$). For each subset size $r$, we average the embeddings $\boldsymbol{E} = \frac{1}{r}\Sigma_{i=1}^{r}\boldsymbol{PL}(P_i^S)$, where $P_i^S$ is the $i^{th}$ paraphrase generated for sentence $\boldsymbol{S}$. This incremental sampling allows us to track how sentence-level representational diversity impacts brain prediction accuracy.

E. **Enriched paraphrases:** We generate $\boldsymbol{R} = 70$ paraphrases (Appendix Table 2) that embed broader contextual and inferential content: details that a human might naturally associate with a sentence, even if not explicitly stated (Appendix Tables 14) and 15. We process these enriched paraphrases in the same way as the standard paraphrases described in D.

F. **Sentence-Generated Images:** We use the stable diffusion model Rombach et al. (2022) to generate $M = 100$ images for each sentence $\boldsymbol{S}$, using the sentence itself as the prompt (Appendix Figure 10). While generating textual descriptions from images is relatively well-studied, expressing the full meaning of a sentence in a single image is far more challenging. Sentences often convey abstract concepts or temporally extended events that cannot be captured in one snapshot. To mitigate this, we create a diverse set of images per sentence, each offering a complementary visual perspective that together approximate the sentence's semantics. From these images we extract embeddings from the penultimate layer of a state-of-the-art vision model $\{\boldsymbol{VM}(I_i^S)\}_{i=1}^{M}$. We then average a subset of $m \in \{5, 10, 15, ., 100\}$ out of $M$ embeddings to form a single representation $E = \frac{1}{m}\Sigma_{i=1}^{m}\boldsymbol{VM}(I_i^S)$, where $I_i^S$ is the $i^{th}$ image generated for sentence $\boldsymbol{S}$.

G. **Content-word images:** For each content word $\{c_j\}_{j=1}^{C^S}$ in sentence $\boldsymbol{S}$, we use the stable diffusion model to generate $M$ images (Appendix Figure 12), from which we extract embeddings from the penultimate layer of a state-of-the-art vision model $\{\boldsymbol{VM}(I_k^{c_j})\}_{k=1}^{M}$. We first average a selected subset of $m$ embeddings to obtain a single representation for each content word, and then average across all content word representations to produce a sentence-level embedding $\boldsymbol{E} = \frac{1}{C^S}\Sigma_{c_j}\frac{1}{m}\Sigma_{k=1}^{m}\boldsymbol{VM}(I_k^{c_j})$, where $I_k^{c_j}$ is the $k^{th}$ image generated for content word $c_j$.

H **Header images:** We generated $M$ images of the same header word (Appendix Figure 11), get vision model embeddings of them, and average a subset $m$ of these images to get an input representation $\boldsymbol{E} = \frac{1}{m}\Sigma_{i=1}^{m}\boldsymbol{VM}(I_i^H)$ to predict $\boldsymbol{Y} = \frac{1}{K}\Sigma_{j=1}^{K}\boldsymbol{Y}_{S_j}$ as described above, where $I_i^H$ is the $i^{th}$ image generated for header word $\boldsymbol{H}$.

We used a wide range of vision models spanning diverse architectures and training objectives He et al. (2016); Liu et al. (2022; 2021); Dosovitskiy et al. (2020); Caron et al. (2021); Radford et al. (2021), and language models spanning encoder–decoder and decoder-only architectures, with causal and non-causal variants Song et al. (2020); Devlin et al. (2019); Radford et al. (2019); Team et al. (2025); Dubey et al. (2024) (detailed descriptions of all models are provided in Appendix Sections A.3 and A.4). We use CLIP strictly as a baseline to assess how much explicit image–text supervision improves prediction. Our core scientific conclusions are derived entirely from unimodal vision models, not from CLIP. All representations from the visual models were extracted from their penul-

timate layers, while embeddings from the language models were computed by averaging the token representations from the final hidden state.

## 3 RESULTS

### 3.1 VISUAL MODELS CAPTURE MEANING IN THE LANGUAGE CORTEX

#### 3.1.1 SENTENCE-LEVEL COMPARISON

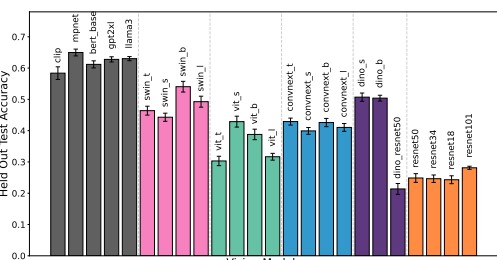
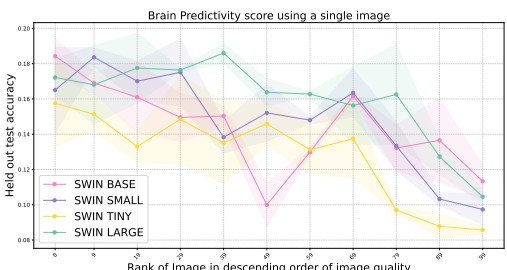

Figure 3: Comparison of performance in predicting language cortex activity between LLM embeddings of the original linguistic stimuli from CSD (2025) dataset (first 5 bars) and single image vision model embedding of their original COCO visual counterparts (remaining bars).

Figure 4: Performance of SWIN vision models in predicting language cortex activity using a single image, with experiments repeated across images sorted in order of decreasing quality (defined as the cosine similarity between the sentence and the image's CLIP embeddings).

First, we compare how vision and language models predict brain responses in the language cortex. Using LLM embeddings of each full sentence and vision-model embeddings of their corresponding generated images (A vs F), we train encoding models to predict cortical activity. With only a single image per sentence, language-based models outperform vision-based ones, though vision embeddings also demonstrate meaningful predictive power for language cortex activity, with some models (such as SWIN transformers) achieving competitive performance. This pattern emerges both in the Pereira (2018) dataset using images generated from sentence prompts (Figure 2) and in the CSD (2025) dataset using original COCO images that correspond to the caption stimuli (Figure 3). As we increase the number of generated images per sentence and average their embeddings, the performance of vision models improves substantially, sometimes even surpassing certain language models (see Pereira (2018) results in Figure 2, Tuckute (2024) results in Appendix Figures 17, and CSD (2025) in Appendix Figure 18; these trends are also consistent across individual subjects: Appendix Figures 28, 29; cross modal overlap in Appendix Figure 36). Further statistical analysis is provided in Appendix section A.6. We also find that unimodal vision models such as Swin sometimes outperform the CLIP vision encoder in predicting language-cortex responses even though CLIP's multimodal training should, in principle, favor it. This likely reflects the fact that CLIP's contrastive objective compresses linguistic dimensions that matter for the brain (CLIP's own text encoder is among the weakest language-model baselines), whereas unimodal vision models preserve broader high-level conceptual structure that aligns more closely with language-cortex semantics.

This cross-modal success demonstrates that language cortex responses can be predicted using representations from visual content that preserves the semantic meaning. While individual images may miss certain semantic nuances, aggregating multiple visual perspectives systematically improves neural prediction accuracy. The robustness of this representational averaging effect, where performance gains from multiple images occur consistently across diverse vision architectures and training objectives highlights that this is a fundamental property of how semantic content can be distilled from varied visual exemplars, even though baseline performance levels differ across model families.

A critical nuance in our analysis is in the quality of the generated images, which is computed as the cosine similarity between CLIP-based embeddings of each image and its corresponding sentence. We use this similarity as a proxy for semantic alignment. Despite being generated from the same sentence, images vary widely in how accurately they capture the intended meaning, as diffusion models are not perfect and often produce off-topic or visually noisy samples (Appendix Figure 13).

Further analysis showing that the generated samples contain meaningful structure rather than noise is provided in Appendix Section A.7.

Currently, all generated images are treated equally during the averaging process. However, only a subset of these images truly reflect the core content of the sentence, while others may diverge significantly. Our analysis shows that this variance directly impacts model performance: the predictive performance of models using single image embeddings declines systematically with decreasing image quality (Figure 4), demonstrating that low quality images fail to encode the critical semantic information present in the original sentence, leading to poorer alignment with brain responses.

We tested whether sorting images by semantic quality yielded more interpretable performance trajectories. When we incrementally include images in descending order of CLIP-based semantic similarity, we observe a sharp rise in accuracy as high-quality images are added, followed by a plateau, and eventually a decline as lower-quality images introduce noise (Figure 5). This contrasts with random ordering, where prediction accuracy is consistently lower and continually increases with exemplar averaging, confirming that image quality systematically affects neural prediction accuracy.

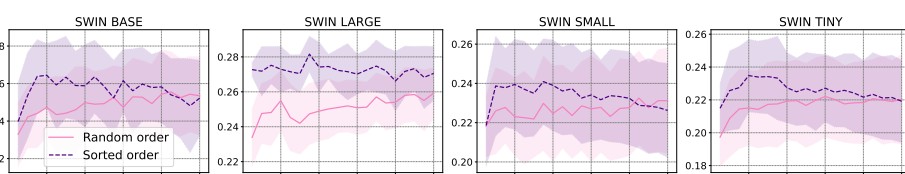

Figure 5: Pereira (2018) Dataset - Comparison of vision model performance in predicting language cortex activity using multiple images, with images ordered randomly versus in order of decreasing quality. Averaging more images helps in the case of random ordering, consistent with averaging the noise from non-ideal images. Adding more images eventually hurts in the case of ordered images, where eventually, less useful images are being incorporated.

### 3.1.2 CONTENT-WORD COMPARISON

Next, we tested whether the core semantic elements of sentences could predict language cortex responses when stripped of their full compositional structure. We evaluated the capacity of vision and language models to predict language-cortex responses by focusing on content words within each sentence—primarily nouns and main verbs that carry the essential semantic content (examples in Appendix Figure 12). This approach reduces sentences to their key conceptual building blocks while removing grammatical structure, function words, and compositional relationships. For each content word, we extracted embeddings from language models, and for the corresponding generated images produced by Stable Diffusion, we obtained visual embeddings using vision networks (featurization B vs G) (Figure 6). Even with a small number of content-word-based images, vision models achieved performance levels comparable to, and in most cases surpassing, those of certain language models such as BERT-Base and MPNet. We also observed a consistent upward trend in accuracy with increasing numbers of images, suggesting that aggregating multiple visual representations of individual concepts captures semantic nuances that contribute to language cortex responses.

These experiments reveal that when we reduce sentences to their core conceptual content, vision and language models achieve comparable predictive power for language cortex responses. The fact that vision model embeddings of images of individual words like 'boy' and 'pancakes' can predict brain responses nearly as well as linguistic representations is particularly striking given that these vision models were trained purely on natural image statistics without language supervision. This suggests that the meaning component of language cortex responses taps into conceptual representations that can be accessed through the statistical structure of visual experience alone, indicating that semantic processing in the language cortex may be grounded in modality-independent principles that both linguistic and visual systems can converge upon. However, the remaining performance differences between content words and full sentences (compare Figures 6 and 2) highlight that compositional structure, grammatical relationships, and contextual integration also contribute meaningfully to language cortex responses, though perhaps to a lesser degree than the core conceptual content.

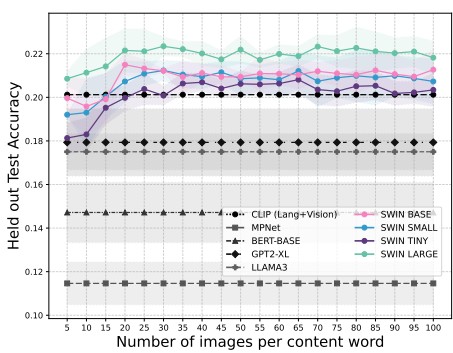 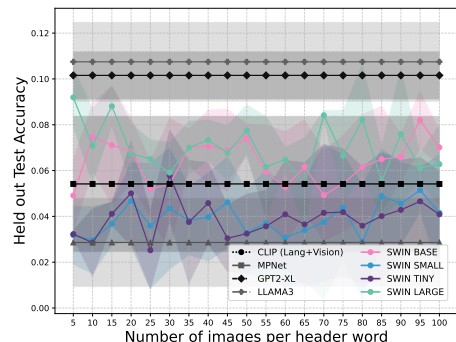

Figure 6: Comparison of performance in predicting language cortex activity between LLM embeddings of the content-word linguistic stimuli and SWIN model embeddings of their visual counterparts. Performance of SWIN vision models (which are trained without language supervision) increases with the number of images, surpassing many of the language models.

Figure 7: Comparison of performance at predicting language cortex activity between LLM embeddings of the header word groups from the Pereira (2018) dataset and SWIN model embeddings of their visual counterparts. For both language and vision models, the performance is worse than when looking at content-words or the original sentence.

### 3.1.3 HEADER-WORD COMPARISON

We now compare LLM embeddings of header words with vision model embeddings of corresponding images to predict averaged brain responses to sentence groups sharing the same topic (C vs H). Vision and language models achieve remarkably comparable performance levels, though both perform substantially lower than when using original sentences or content words (Figure 7). The averaging effect with increasing numbers of images is consistent in smaller Swin models but modest, possibly because individual header words are already concrete and semantically focused enough that a few images capture their core meaning. This experiment represents the most abstract test of cross-modal prediction: using single thematic words (e.g., 'toaster', 'beekeeping') to predict language cortex responses to entire passages on those topics. The near-equivalence between vision and language models at this level suggests that when semantic content is highly abstracted, modality differences become minimal. However, the overall lower accuracy indicates that much of the predictive power for language cortex responses comes from the richer semantic and compositional information present in full sentences and individual concepts, rather than abstract thematic categories alone.

### 3.2 PARAPHRASES

#### 3.2.1 COMPARING ORIGINAL SENTENCES WITH PARAPHRASES

So far, we examined the potential of cross-modal representations for modeling the language cortex. Here, we turn to alternative representations within the same modality, comparing the modeling performance of LLM embeddings of the original sentence with the mean embeddings of its paraphrases (A vs D). In the Pereira (2018) dataset, paraphrase embeddings alone yielded reasonable prediction accuracy. While averaging over a small number of paraphrases ($\leq 5$) underperformed relative to the original sentence embeddings (Figure 8), performance improved steadily as more paraphrases were incorporated (pink curve), at times surpassing the original sentence baseline (grey curve). This benefit was weaker in the Tuckute (2024) dataset (Appendix Figure 20), likely for two reasons: (i) responses in this dataset were averaged across voxels, which eliminated voxel-specific information that may carry important signal, and (ii) Pereira (2018) paragraphs are longer, richer and more content-diverse, making paraphrases more likely to be diverse, whereas Tuckute (2024) sentences are simpler and more abstract, yielding paraphrases with limited variation (Appendix Tables 14, 15).

This pattern parallels our vision-based experiments, where a single image was less predictive than the original sentence but prediction accuracy improved as multiple visual samples were averaged. As in the vision case, alternative modalities or representational forms show greater potential for modeling the language cortex when the original stimuli presented to subjects are information-rich and

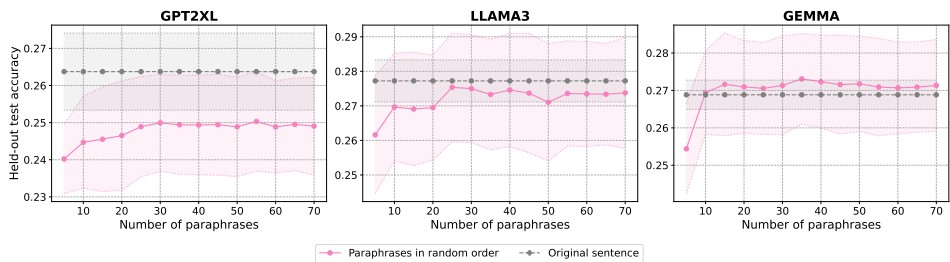

Figure 8: Pereira (2018) dataset - Comparison of the performance in predicting language cortex activity by LLM embeddings of the original linguistic stimuli presented to subjects with averaged embeddings of generated paraphrases. Averaging embeddings across more paraphrases steadily improves prediction accuracy, in some cases matching or surpassing the original sentence.

concrete. Similarly, when we sort the paraphrases based on their semantic similarity to the original sentence (measured using the cosine similarity in the CLIP language encoder space) and incrementally add them in descending order of similarity (Appendix Figures 21 and 22), we observe a similar trend: accuracy improves rapidly at first and then saturates (green plot). This suggests diminishing returns as lower quality or less semantically similar paraphrases are added, highlighting that not all paraphrases contribute equally, and the quality of information matters as much as quantity.

Averaging paraphrase embeddings narrows, but typically does not erase the gap to the original sentence; it exceeds the original mainly with stronger LLMs (Gemma3) and for semantically rich stimuli (e.g., Pereira (2018)). This pattern suggests two components in the language cortex signal: (i) a content-dominant component recoverable from many surface realizations, and (ii) a form-sensitive residual captured best by the original sentence.

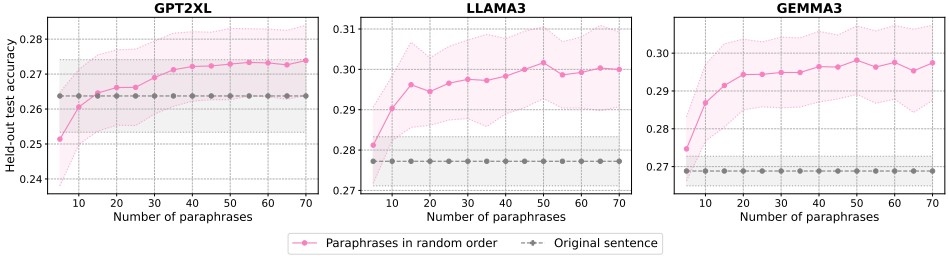

Figure 9: Pereira (2018) Dataset - Comparison of LLM embeddings of the original linguistic stimuli presented to subjects with averaged embeddings of generated paraphrases with additional context. Averaging embeddings of paraphrases enriched with contextual information yields higher prediction accuracy than the original sentence embeddings.

### 3.2.2 COMPARING ORIGINAL SENTENCES WITH COMMONSENSE-ENRICHED PARAPHRASES

Our earlier experiments hinted that adding supplemental information to a sentence, beyond its original form often enhances our ability to model brain responses. However, the generated paraphrases were semantically near-identical to the original sentence, merely differing in surface structure. Thus, the added information was more a stylistic variation than a substantive expansion of meaning.

This led us to ask a deeper question: can paraphrases enriched with new commonsense context, while still preserving the core semantics of the original sentence further improve brain predictivity (featurization A vs E)? In this next set of experiments (illustrated in Figure 9, Appendix figure 31), we intentionally moved beyond superficial rewording and generated paraphrases that embedded broader contextual and inferential content: details that a human might naturally associate with the sentence, even if not explicitly stated. These enriched paraphrases preserve the original intent, but vary more significantly in both form and informational density. More broadly, we were interested in whether adding such rich context could compensate for or even outweigh the importance of original form or modality when modeling responses in the language cortex (Appendix Table 14).

We found that the effectiveness of this approach depended strongly on the nature of the dataset. In the Pereira (2018) dataset, whose sentences are typically rich, concrete, and highly visualizable, it was relatively easy to generate extended paraphrases with coherent, plausible context. In these cases, even a small number of enriched paraphrases substantially outperformed the original sentences in modeling brain activity (pink curve in Figure 9), and performance continued to improve without plateauing. This suggests that additional meaningful information, particularly information the brain may implicitly infer can have a strong additive effect on neural predictivity. We next asked whether combining the original sentence with these enriched paraphrases could further improve model performance, reasoning that the specific linguistic form of the original sentence might add information beyond paraphrased content (Appendix Figure 26). When we concatenated the original sentence embedding with the averaged paraphrase embedding, we did not see as major an improvement as we had seen when we used paraphrases without additional context. For older or less powerful models, such as GPT-2 XL, concatenation yielded a modest boost relative to paraphrases alone, likely because these models benefit from the precise structure of the original sentence combined with the additional semantic variation from the paraphrases. However, for more powerful LLMs, such as Llama 3 and Gemma 3, the improvement was negligible, and in some cases the paraphrases alone performed slightly better than the concatenated representation. These stronger models already encode rich, contextually robust representations of the original sentence, so the explicit addition of the original embedding contributes little new information.

By contrast, for the Tuckute (2024) dataset this strategy was less effective. These sentences are short (6 words long) and some of them abstract, making it difficult to generate paraphrases that are both semantically coherent and genuinely informative. Many paraphrases introduced noise or drifted from the original meaning (Appendix Table 15), so the enriched paraphrases alone failed to outperform the original sentences (Appendix Figure 25). In this setting, the exact linguistic form of the stimulus carried more predictive power than additional semantic content. As a result, appending the original sentence by concatenation led to performance gains, although it does not always beat the prediction accuracy when the embedding of the original sentence is used. (Appendix Figure 27).

These results suggest that predictive accuracy in the language cortex is shaped strongly by the informativeness of semantic content. When added information is meaningful and consistent with the brain's associative priors, it enhances predictivity. Thus, enriching inputs with inferential context can provide substantial gains beyond what is available from the original linguistic form alone. Appendix Section A.6 provides additional statistical analyses of our results. Appendix Section A.7 further reports ablation studies demonstrating that the paraphrases encode semantically relevant information rather than noise, and that the improvement in modeling performance stems from this enriched information content rather than from incidental increases in sentence length.

## 4 DISCUSSION

In this work, we investigated whether the human language cortex can be modeled by stimuli that differ from the original linguistic input in modality or form while preserving semantic content. Our central aim was to examine the limits of representational flexibility, that is, the extent to which neural responses to linguistic stimuli can be accurately predicted using representations derived from alternative inputs that vary dramatically in surface form (paraphrases), modality (visual vs. linguistic), and information density (enriched vs. original content).

These analyses characterize the degree of abstraction in semantic representations within the language cortex, demonstrating that model-predictable variance can be captured by representations that vary substantially in form and modality, revealing more abstract structure than previously shown using standard linguistic models alone. This conclusion emerges from three key observations: vision model embeddings can predict language cortex responses when aggregated across multiple images, paraphrase embeddings show similar predictive power when averaged, and semantically enriched paraphrases can exceed the predictive accuracy of original sentences.

The success of cross-modal prediction using visual representations to model language cortex responses is particularly striking given that the language cortex was not exposed to any visual input during the original experiments. This suggests that meaning in the language cortex is encoded in a format that transcends specific sensory modalities. Further, the effectiveness of representational averaging across examples that share semantic content, whether multiple images or paraphrases,

suggests that this process isolates the shared semantic component by reducing surface-level variability. This finding demonstrates that the meaning component of language cortex responses can be captured even when input content is embedded in highly variable surface forms or accessed through entirely different sensory modalities. Additionally, the finding that enriched paraphrases can exceed original sentence predictions suggests that the language cortex constructs enriched semantic representations that extend far beyond the information literally present in the text, incorporating the contextual associations and commonsense inferences that humans naturally bring to comprehension. Importantly, these results should not be interpreted as suggesting that the language network represents arbitrary extralinguistic facts. Rather, enriched paraphrases serve as a modeling tool: they shift the embedding toward the event-level structure that listeners typically recover during comprehension, allowing the model to better approximate the neural response. This interpretation is fully consistent with prior work showing that the language system builds structured, linguistically mediated representations of events while not storing encyclopedic world knowledge. This finding suggests a core difference in the breadth of semantic associations that brains and language models maintain, with the brain capturing richer and more encompassing representations, perhaps due to the broader range of real-world tasks it learns to perform that go beyond next word prediction.

A parallel line of research has examined cross-modal semantic access in the brain, asking where semantic information can be decoded irrespective of whether it is presented visually or linguistically. For example, Nikolaus et al. (2024) show that certain high-level visual regions support cross-modal identification between images and captions, and earlier decoding studies similarly report category-level generalization across words, pictures, and sounds in some brain areas(Simanova et al., 2014; Shinkareva et al., 2011; Fairhall & Caramazza, 2013). These studies address the localization of modality-general semantic information, identifying brain areas from which a decoder can recover stimulus identity across modalities. In contrast, the present work does not aim to localize where cross-modal semantics resides. Instead, we focus on the representational format of meaning within the language-selective cortex itself, and ask whether its internal semantic structure can be approximated using representational sources that differ markedly from the original linguistic input (e.g., unimodal vision models, paraphrastic variants, enriched paraphrases). This question about the nature of meaning representations within the language system is distinct from prior work centered on where cross-modal semantic information is accessible in the brain.

Several methodological constraints should be acknowledged. First, our claims characterize the degree of abstraction in model-predictable cortical variance rather than making absolute assertions about form-independence in the language cortex's representations of meaning. The success of diverse representational sources (vision models, paraphrases, enriched content) demonstrates that the components of neural responses currently accessible to computational models reflect relatively abstract semantic structure. However, the residual advantage of original sentences indicates that form-specific information contributes to neural responses, even if less accessible to current embeddings. Our correlation-based encoding approach reveals which representational spaces align with language cortex activity but cannot establish whether the brain's meaning representations are genuinely independent of linguistic form, nor can it reveal how these representations support comprehension or behavior. Second, our visual generation relied on current diffusion models, which may not optimally capture the visual content most relevant to language processing. Future work could explore whether more sophisticated multimodal models or human-generated images yield different patterns. Third, our commonsense enrichment was generated algorithmically and may not reflect the specific knowledge integration processes that occur during natural reading or listening, where representations evolve on a word-by-word basis. Lastly, the fMRI datasets used in this study have inherently slow temporal resolution, providing only coarse snapshots of neural activity averaged over several seconds rather than millisecond-level dynamics. As in all work on the language network, this limits the temporal precision of the representational inferences one can draw, and future work using higher-temporal-resolution modalities (e.g., intracranial recordings) will be valuable.

Despite these limitations, our work demonstrates how powerful generative models can be leveraged to create alternative input types that preserve semantic content while varying in modality (visual vs. linguistic) and surface form (paraphrases), enabling cross-modal and cross-form neural prediction. Through averaging model representations of multiple generated variants, we can isolate the shared meaning components that drive neural responses. This methodological approach opens new avenues for understanding how intelligent systems, both biological and artificial, extract and represent meaning from experience that can be captured across diverse sensory modalities and surface forms.

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

# A APPENDIX

## A.1 DATASETS

The following datasets were used in our study:

1. Pereira et al. (2018), who recorded brain responses from 16 native English speakers as they silently read short, syntactically and semantically diverse passages. The study comprised three experiments: (1) isolated words and simple phrases to probe basic lexical–semantic representations, (2) 384 full sentences grouped into 96 short narratives on varied concrete topics such as professions, clothing, birds, musical instruments, natural disasters, and crimes, and (3) an additional 243 full sentences organized into 72 narratives of similar thematic breadth. For our analyses, we focus on the five participants who completed both Experiments 2 and 3, using only these two experiments because they provide full-length sentence stimuli: 627 unique sentences in total, suitable for modeling rich compositional semantics.

## Pereira 2018 Dataset

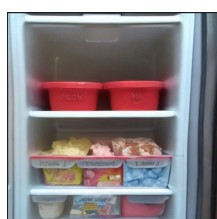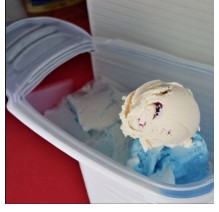

**Then I put the container of half-frozen ice cream into the freezer to harden.**

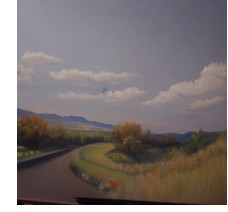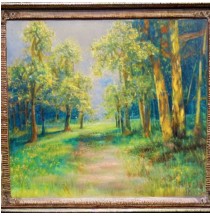

**As a painter, I learned to focus less on the actual scene and more on the painting itself.**

## Tuckute 2024 Dataset

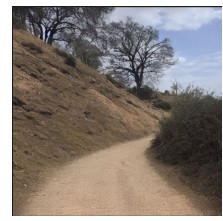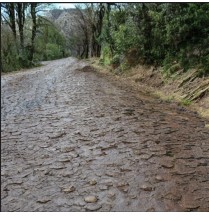

**The weather was warm and dry.**

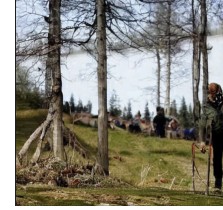

**Nordmann considered it all very carefully.**

## Caption Scene Dataset 2025

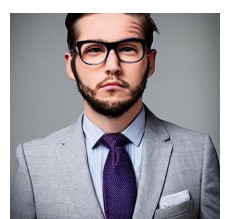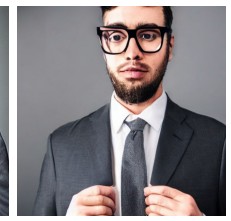

**This man is wearing a suit, tie, and glasses.**

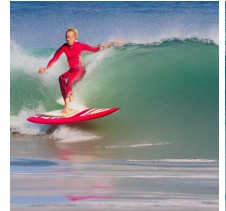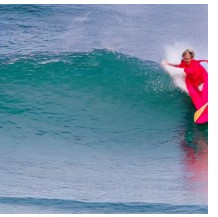

**A woman wearing a red shirt is surfing at sea**

Figure 10: Examples of sentences used, and images generated for these sentences using Stable Diffusion

2. Tuckute et al. (2024b), who recorded brain responses from five native English speakers as they read 1,000 six-word sentences in an event-related fMRI design. The sentences were sampled from text corpora to maximize both semantic coverage and stylistic diversity. Each sentence was presented individually (2 s on screen, 4 s inter-stimulus interval), with runs of 50 sentences and short fixation blocks interleaved. Participants were instructed to read attentively and think about the meaning of each sentence, and to encourage engagement, they were informed that a brief memory task would follow the scan. Because each stimulus was presented to participants only once, we trained the encoding models to predict the aver-

**Bed**

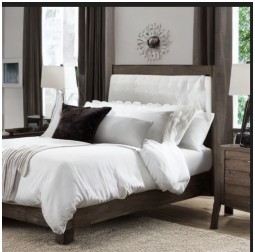

1. A bed is a piece of furniture used as a place to sleep or have sex in.
2. A bed is made of a mattress and a box spring, plus sheets, pillows and covers.
3. In waterbeds the mattress is filled with water, and in airbeds it is filled with air.
4. Another type of bed is the hammock, a fabric sling suspended above the ground.

**Toaster**

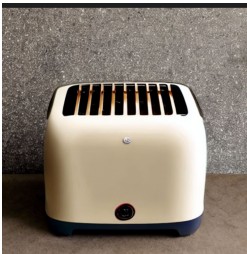

1. A toaster is a kitchen appliance for toasting bread using heating elements.
2. Pop-up toasters toast the bread placed in a slot and eject it once done.
3. Toaster ovens have a door on the side, a tray within and temperature control.
4. Toaster ovens allow bread to be cooked with toppings like garlic or cheese.

Figure 11: Examples of header words for a group of sentences in Pereira (2018) dataset, and images generated for these header words using Stable Diffusion

**Pereira 2018 Dataset**

Early crews were all young men, but astronauts now are much more diverse.

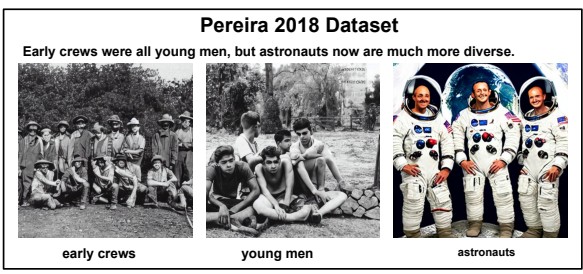

early crews          young men          astronauts

**Tuckute 2024 Dataset**

We love rich tourists on vacation.

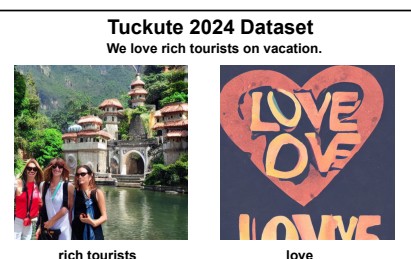

rich tourists          love

Figure 12: Examples of content words/phrases in a sentence, and images generated for these content words/phrases using Stable Diffusion

age response across all voxels in the functionally-defined five language fROIs. This dataset provides brain responses to a semantically and stylistically diverse set of decontextualized sentences, well suited for modeling sentence-level representations.

Note that the sentences in this dataset are relatively abstract and simpler compared to those in the Pereira (2018) dataset. As a result, many sentences lack clearly identifiable content words (see Section 2). For this reason, we did not perform the content-word experiments on this dataset.

3. Li et al. (2025) (Caption Scene Dataset) comprises fMRI data from eight participants performing a semantic matching task: each participant first read a Chinese caption and then viewed a corresponding MS-COCO image Lin et al. (2014) to decide whether the text and image matched semantically. For our analyses, we focused on the four subjects with the highest signal-to-noise ratios. Because the image was shown only after the caption, the neural responses during caption reading are uncontaminated by visual input. We therefore analyze only these "pure language" responses from the caption-reading phase. The full dataset contains 9,375 unique captions and 9,494 images (18,893 total stimuli). From these, we use the 983 caption stimuli that were presented to all four selected subjects. The accuracies reported on this dataset are noise-normalized, with noise ceiling computed following the NSD procedure Allen et al. (2022). In our case, given only two repetitions for

**Bees crawl across his bare arms and hands, but they don't sting, because they're gentle**

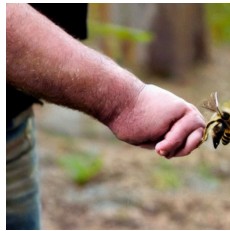 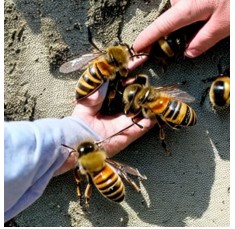 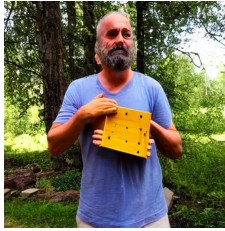

**In developed countries, blindness is more likely to be caused by genetic problems.**

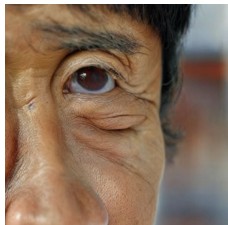 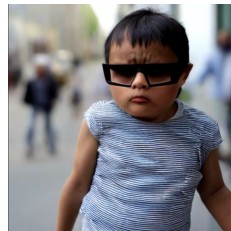

Figure 13: Examples of images generated for a given sentence, ranked in descending order of quality. Image quality is measured by the cosine similarity between the CLIP embeddings of the original sentence and the generated images.

| Original Sentence | Paraphrases without any additional context | Paraphrases with commonsense context |
|---|---|---|
| Theft is a crime and can be punishable with jail or fines. | Illegally taking items is a criminal offense, resulting in prison or fines. | The unlawful appropriation of another's property is theft, a criminal offense that is subject to penalties including jail or financial sanctions imposed by the legal system. |
| | Committing theft incurs punishment such as jail or a fine. | The unauthorized taking of another's property, known as theft, is against the law and can result in penalties such as serving time in prison or paying monetary fines. |
| | Taking items illegally results in legal consequences: jail or fines. | The unlawful appropriation of property is theft, a criminal offense that is subject to penalties including jail or financial sanctions. |
| Many bees have stingers and can attack if they feel threatened. | Bees will sting to protect their colony when threatened. | The stinging apparatus of a bee is its main deterrent against predators or disturbance when it feels threatened. |
| | Bees can deliver a sting when they are feeling threatened. | The primary purpose of a bee's stinger is self-protection and the defense of its colony from harm. |
| | Stinging is how bees react to feeling unsafe or attacked. | Be careful around bees, especially near flowers or potential hive sites, as they can sting if they feel their space is invaded. |

Figure 14: Examples of paraphrases generated for sentences in the Pereira (2018) dataset

each stimulus, the noise ceiling is equivalent to split-half reliability. Note that since the captions in the Captions Scene Dataset are written in Chinese, we do not apply our linguistic modulation analyses, such as isolating content words or generating paraphrases to this dataset.

## A.2 TRAINING DETAILS

We used ridge regression–based encoding models to predict activity in the language-selective cortex. For each stimulus, the model outputs a vector of length $N$, where $N$ denotes the total number of voxels whose responses are being predicted. Since multiple participants were available across each dataset, we concatenated their voxel responses into a single target vector. For the Tuckute (2024)

| Original Sentence | Paraphrases without any additional context | Paraphrases with commonsense context |
|---|---|---|
| Do any stand out in particular? | Which feature has the most impact? | Which features in the photograph do you find most remarkable or noteworthy? |
| | What's your primary focus in this shot? | What features in the scene are most distinguishable or visually prominent? |
| | See anything that sparks curiosity? | Does anything seem particularly sharp or in focus compared to the rest? |
| The hesitant rhythm of the band. | The band's rhythm feels reluctant. | Playing live, the band exhibited a hesitant rhythm, possibly due to opening night jitters. |
| | Uncertainty marks the band's rhythm. | The band played with a hesitant rhythm, unsure of the upcoming key change. |
| | The band's rhythm is marked by hesitation. | The rhythm played by the band was hesitant, making it hard to tap your foot along. |

Figure 15: Examples of paraphrases generated for sentences in the Tuckute (2024) dataset

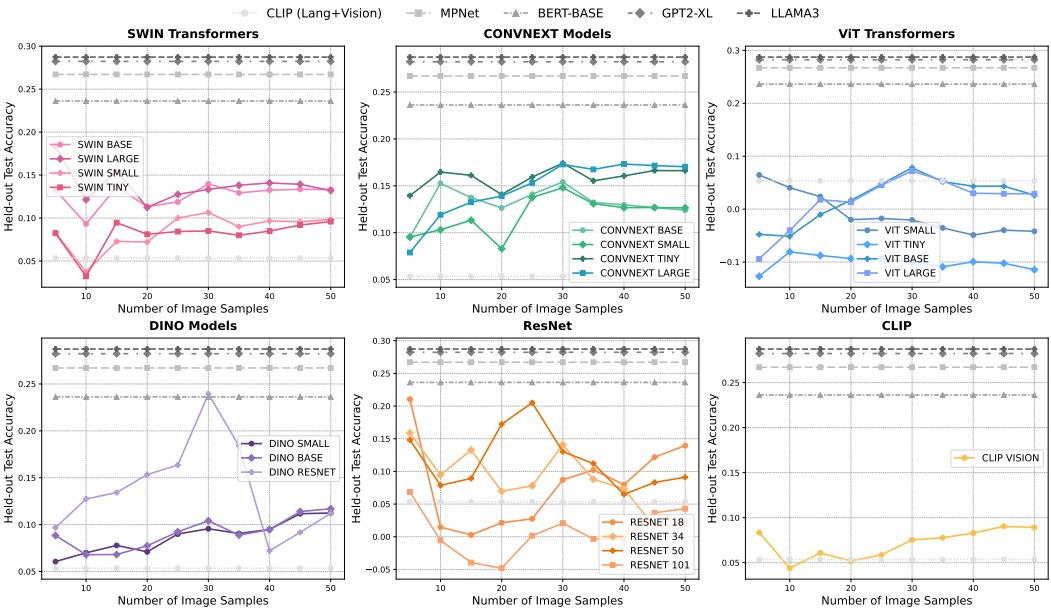

Figure 16: Performance comparison between LLM embeddings of the original linguistic stimuli from the Tuckute (2024) dataset and vision model embeddings of their corresponding visual counterparts.

dataset, only single-trial responses were provided; to reduce noise, we averaged the $N$ voxels into a single scalar response per stimulus.

To obtain robust estimates of model performance, we generated multiple outer train–test splits for each dataset (three splits for Pereira (2018), five splits for all others) and reported results averaged across the test splits. For the Pereira (2018) dataset, we designed these splits to ensure a fair evaluation, since having sentences from the same paragraph appear in both training and test sets could allow the model to exploit shared context Kauf et al. (2024); Feghhi et al. (2024). To assess whether the observed trends hold independently of contextual overlap and temporal autocorrelation, we created two random splits and one targeted split in which the test set contained only the first sentence from each of 63 paragraphs. All reported results on Pereira (2018) were averaged across these three

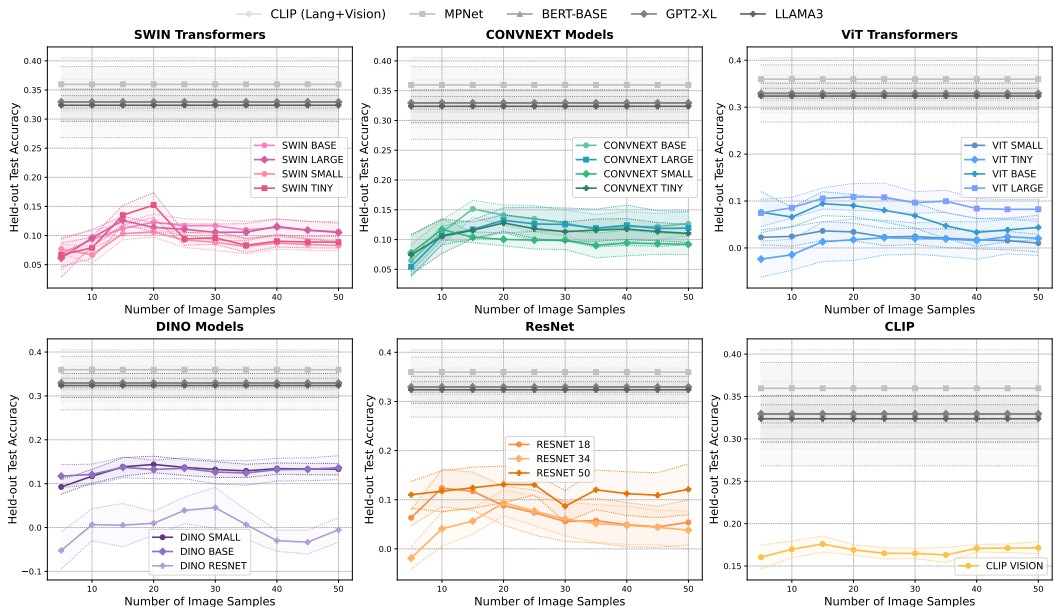

Figure 17: Evaluating LLM embeddings of Tuckute (2024) sentences versus vision-model embeddings of their visual counterparts, using only sentences with generated images of sufficient quality (mean CLIP score $\geq 0.25$)

test splits. Results on the Tuckute (2024) and CSD (2025) datasets similarly reflect averages over five test splits.

For the Pereira (2018) dataset, voxel responses occasionally contained `NaN` measurements, distributed sporadically across stimuli. No voxel was consistently missing across all examples, and discarding affected voxels would cause substantial data loss. To retain the full voxel set, we trained the encoding model via backpropagation by further splitting each outer train split into an inner train–validation split and optimizing a combined loss consisting of mean squared error and correlation between predicted and observed voxel responses. Hyperparameters were then tuned within each outer train split using cross-validation over a grid of ridge penalties $\lambda \in (0.0, 0.1, 0.01, 0.001, 0.0001)$.

For the remaining datasets, voxel responses were estimated using the closed-form ridge regression solution, with $\lambda$ selected by inner k-fold cross-validation performed only on the outer train split (leave-one-out cross-validation with $K = M$, where $M$ is the number of samples, for Tuckute (2024); $k = 10$ for CSD (2025)). Test performance for all datasets was quantified as the Pearson correlation between predicted and observed voxel responses on the held-out test set.

## A.3 LANGUAGE MODELS USED FOR EVALUATION

1. **CLIP Language Encoder** Radford et al. (2021): The text branch of CLIP, trained jointly with its vision counterpart using a contrastive image–text objective. It produces rich sentence-level embeddings aligned to visual concepts, enabling zero-shot cross-modal tasks.

2. **MPNET** Song et al. (2020): A Transformer-based language model that combines masked language modeling with permutation-based training, allowing it to capture bidirectional context and sequential dependency simultaneously.

3. **BERT Base** Devlin et al. (2019): A bidirectional Transformer pretrained with masked language modeling, providing contextual word and sentence embeddings.

4. **GPT-2 XL** Radford et al. (2019): An autoregressive Transformer with 1.5 B parameters, trained to predict the next token in large-scale web text.

5. **Gemma-3** Team et al. (2025): Google's latest open-weight decoder-only Transformer with grouped-query attention and alternating local/global layers for efficient long-context reasoning (up to 128K tokens). Larger variants add a SigLIP vision encoder for multimodal inputs and support quantization for lightweight deployment.

6. **LLaMA-3** Dubey et al. (2024): Meta's third-generation decoder-only Transformer featuring grouped-query attention and efficient scaling.

## A.4 VISION MODELS USED FOR EVALUATION

We use a broad range of vision models spanning diverse architectures and training objectives to demonstrate that our findings are broadly reproducible rather than specific to any single model.

1. **ResNet family** He et al. (2016) (ResNet-50, ResNet-34, ResNet-18, ResNet-101): Deep residual networks that use skip connections to mitigate vanishing gradients, enabling very deep convolutional architectures. The pretrained weights, obtained from PyTorch, were trained on the ImageNet-1K dataset.

2. **ConvNeXt family** (ConvNeXt Small, Tiny, Base, Large) Liu et al. (2022): A modernized CNN design that incorporates architectural ideas from Transformers, such as large kernels and inverted bottlenecks. The pretrained weights, obtained from PyTorch, were trained on the ImageNet-1K dataset.

3. **Swin Transformers** Liu et al. (2021) (Swin Small, Tiny, Base, Large): Hierarchical vision Transformers that process images using shifted windows, providing linear computational complexity with respect to image size. The pretrained weights, obtained from Huggingface, were trained on the ImageNet-1K dataset.

4. **Vision Transformers (ViTs)** (ViT Small, Tiny, Base, Large) Dosovitskiy et al. (2020): Pure Transformer architectures that treat images as sequences of non-overlapping patches, capturing global context through self-attention. The pretrained weights, obtained from timm, were trained on the ImageNet-1K dataset.

5. **DINO models** (DINO Small, Base, ResNet-50) Caron et al. (2021): Self-supervised representations learned via knowledge distillation, producing strong, transferable visual features without labeled data and supporting both ViT and ResNet backbones. The pretrained weights, obtained from torch hub, were trained on the ImageNet-1K dataset.

6. **CLIP Vision Encoder** Radford et al. (2021): The visual branch of CLIP, trained on $\sim$ 400M publicly sourced image–text pairs (WebImageText) with a contrastive objective and released by OpenAI to align images and text in a shared embedding space, enabling zero-shot recognition and robust cross-modal retrieval.

## A.5 EVALUATING VISION- AND LANGUAGE-MODEL PREDICTIONS OF LANGUAGE CORTEX ACTIVITY - TUCKUTE (2024)

Applying the same vision-versus-language modeling framework to the Tuckute (2024) dataset as in Section 3.1.1, we do not observe the clear performance gains with increasing image counts seen for Pereira (2018). Many vision models fail to learn meaningful mappings, as indicated by their negative test-time correlation scores (Figure 16). This likely reflects the more abstract nature of the Tuckute (2024) sentences, which makes generating semantically aligned visuals more challenging (Figure 10). Nevertheless, the performance of vision-based models remains reasonably competitive, indicating that even abstract sentences can evoke visual representations that capture meaningful information, albeit with more noise. Importantly, this observation still aligns with our core hypothesis: even when vision-based representations are noisier, they retain enough semantic signal to contribute meaningfully to neural prediction.

To account for this variability, we filtered the Tuckute (2024) dataset to retain only those images with a CLIP-based quality score above 0.25. Restricting our analysis to these higher-quality samples allowed us to recover the expected performance trend: as the number of relevant images increases, model accuracy improves. However, this trend is less not observed in ViT-based models (Figure 17). One possible explanation is that standard ViT architectures rely on fixed-size, non-overlapping image patches and lack mechanisms for hierarchical feature extraction or localized inductive biases.

As a result, they may be less sensitive to fine-grained variations in image quality or spatial detail, especially in scenarios where the semantic content is subtle or distributed across small regions of the image.

fMRI datasets tailored for multimodal modeling, especially those that align closely with the hypotheses we aim to test are exceptionally rare. In particular, not all datasets contain sentences that are as visually grounded and easily translatable into images as those in the Pereira (2018) dataset. For example, many sentences in the Tuckute (2024) dataset are more abstract in nature, making it substantially more difficult to generate meaningful visual representations. This challenge is evident in the lower quality and occasionally off-topic images produced by the diffusion model, as shown in Appendix Figure 10.

### A.6 EVALUATING VISION- AND LANGUAGE-MODEL PREDICTIONS OF LANGUAGE CORTEX ACTIVITY (CAPTION SCENE DATASET 2025)

Lastly, we compare language-model embeddings of the full original sentence with vision-model embeddings of the corresponding visual scenes using the Caption Scene Dataset (2025), as described in Section 3.1.1. Because participants first read captions describing MS-COCO images, we begin by contrasting the language-model embeddings of these captions with the vision-model embeddings of the original COCO images.

Although the language models better capture brain activity in the language cortex, the vision models achieve performance that is only slightly lower. This reinforces our earlier finding that language cortex prediction remains highly sensitive to the exact linguistic form, giving language models an advantage, while the underlying semantic content can be represented nearly as well through visual modality.

We then extend this analysis by generating multiple synthetic images for each caption with Stable Diffusion and averaging their vision embeddings, following the procedure in Section 3.1.1. The same pattern emerges as in the Pereira (2018) and Tuckute (2024) datasets (Figures 2 and 17): vision-model performance improves as the number of generated samples increases, in some cases approaching that of the language models.

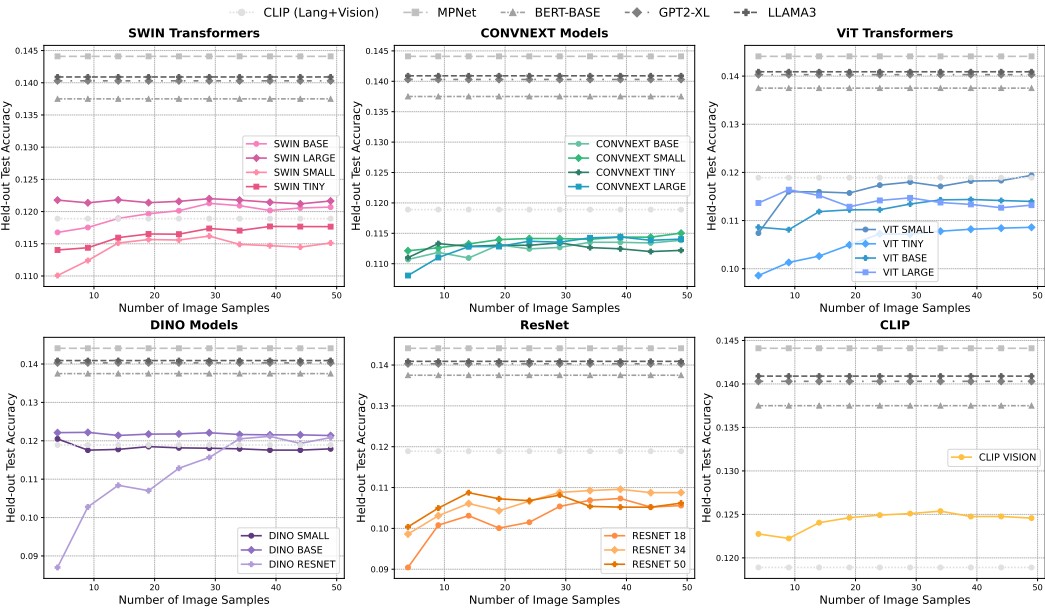

Figure 18: Performance comparison between LLM embeddings of the original linguistic stimuli from the CSD (2025) dataset and vision model embeddings of their corresponding visual counterparts.

**Prompt to generate paraphrases**

```
Prompt:
f"""You are an expert image captioner. I'll show you some existing captions for an
    image, and your task is to generate 70 NEW captions that:
    1. Are similar in style and detail level to the existing captions
    2. Capture the same meaning but with different wording
    3. Are direct, concise descriptions (around 10-15 words each)
    4. Are worded differently from each existing caption and from each other

Here are the existing captions:
{insert all captions text for the image here}

Generate 10 new captions formatted exactly as:
1. [First new caption]
2. [Second new caption]
3. [Third new caption]
4. [Fourth new caption]
5. [Fifth new caption]
6. [Sixth new caption]
7. [Seventh new caption]
8. [Eighth new caption]
9. [Ninth new caption]
10. [Tenth new caption] ... """
```

Table 1: Prompt used for generating paraphrase using Gemini-2.5-Flash.

**Prompt to generate paraphrases with extra context**

```
Prompt:
f"""You are an expert image captioner. I'll show you existing captions for an image,
and your task is to generate 70 NEW captions that:

1. Are similar in style but include more detail than the existing captions
2. Capture the same meaning but with different wording
3. Are worded differently from each existing caption and from each other
4. Contain additional commonsense context to the original caption

Example:
For the sentence 'The boy is eating pancakes for breakfast', some paraphrases
with additional context would be:
  1. The boy is eating pancakes with maple syrup in the morning for breakfast
  2. The boy is sitting at the dining table and having pancakes for breakfast

Here are the existing captions:
{insert all captions text for the image here}

Generate 10 new captions formatted exactly as:
1. [First new caption]
2. [Second new caption]
3. [Third new caption]
4. [Fourth new caption]
5. [Fifth new caption]
6. [Sixth new caption]
7. [Seventh new caption]
8. [Eighth new caption]
9. [Ninth new caption]
10. [Tenth new caption] ... """
```

Table 2: Prompt used for generating paraphrases with additional commonsense context using Gemini-2.5-Flash.

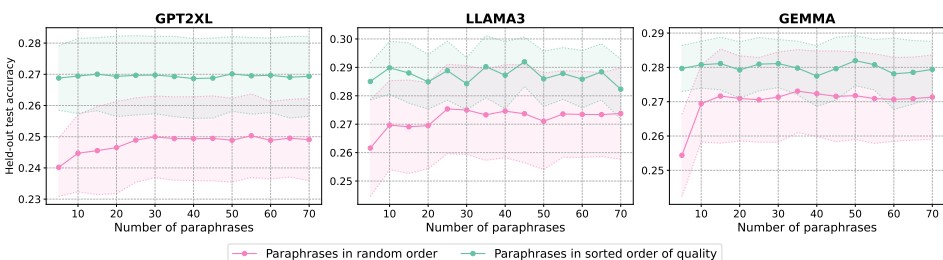

Figure 19: Pereira (2018) dataset - Comparison of averaged LLM embeddings of paraphrases in random order with those arranged in sorted order of semantic similarity to the original sentence. These paraphrases do not have added commonsense context.

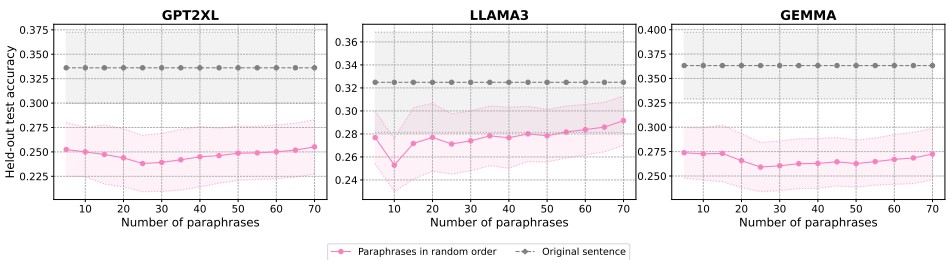

Figure 20: Tuckute (2024) dataset - Comparison of LLM embeddings of the original linguistic stimuli presented to subjects with averaged embeddings of generated paraphrases without additional context.

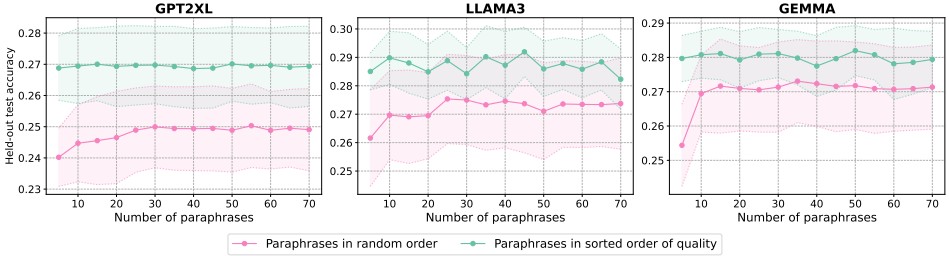

Figure 21: Pereira (2018) dataset - Comparison of averaged LLM embeddings of paraphrases in random order with those arranged in sorted order of semantic similarity to the original sentence. These paraphrases do not have added commonsense context.

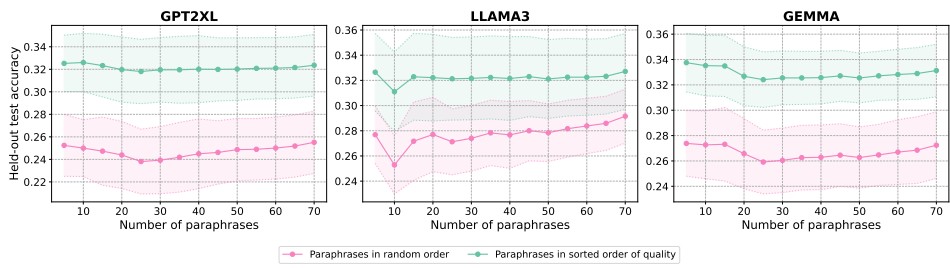

Figure 22: Tuckute (2024) dataset - Comparison of averaged LLM embeddings of paraphrases in random order with those arranged in sorted order of semantic similarity to the original sentence. These paraphrases do not have added commonsense context.

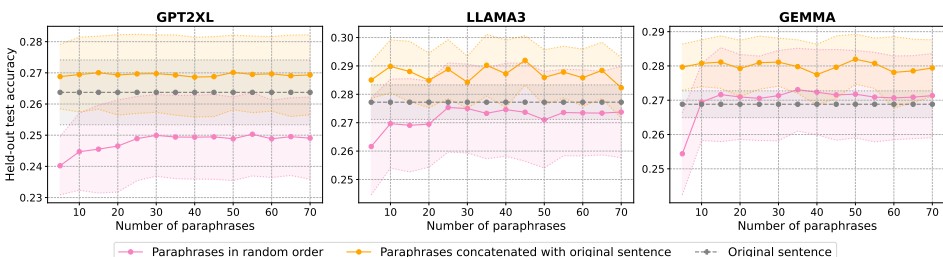

Figure 23: Pereira (2018) dataset - Comparison of averaged LLM embeddings of paraphrases alone with those that are concatenated with the original sentence. These paraphrases do not have added commonsense context.

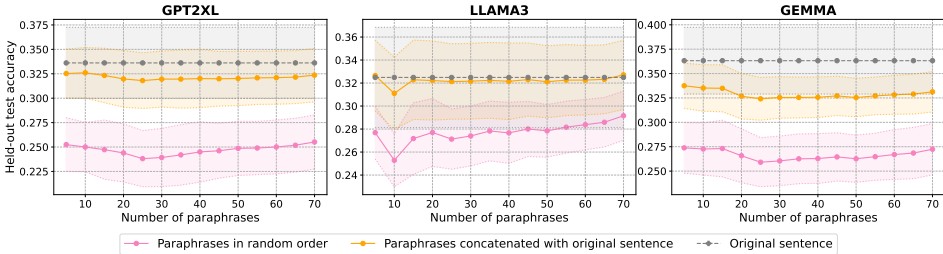

Figure 24: Tuckute (2024) dataset - Comparison of averaged LLM embeddings of paraphrases alone with those that are concatenated with the original sentence. These paraphrases do not have added commonsense context.

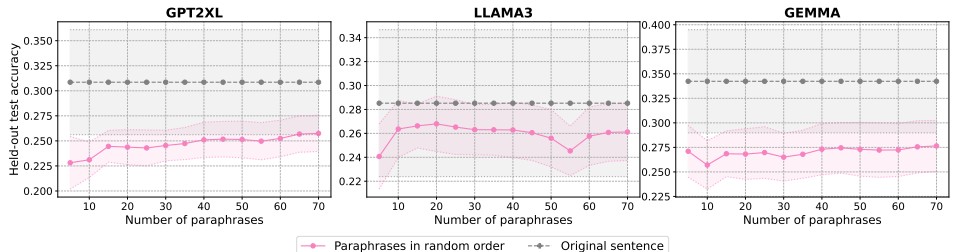

Figure 25: Tuckute (2024) dataset - Comparison of LLM embeddings of the original linguistic stimuli presented to subjects with averaged embeddings of generated paraphrases with additional context.

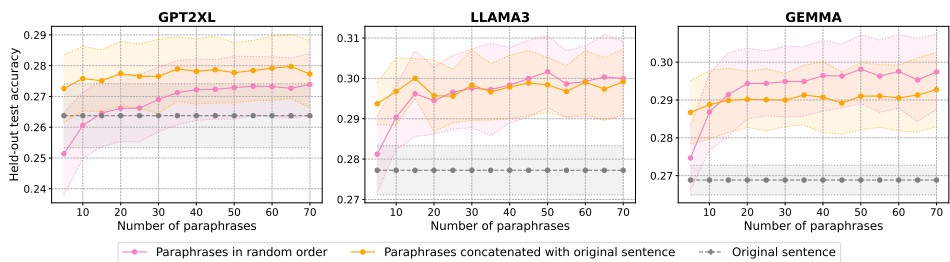

Figure 26: Pereira (2018) dataset - Comparison of averaged LLM embeddings of paraphrases alone with those that are concatenated with the original sentence. These paraphrases have added commonsense context.

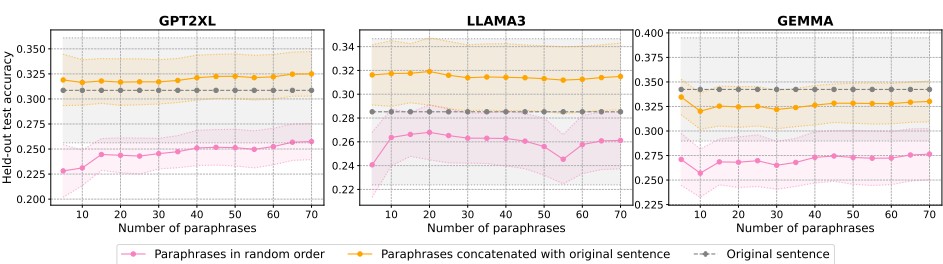

Figure 27: Tuckute (2024) dataset - Comparison of averaged LLM embeddings of paraphrases alone with those that are concatenated with the original sentence. These paraphrases have added common-sense context.

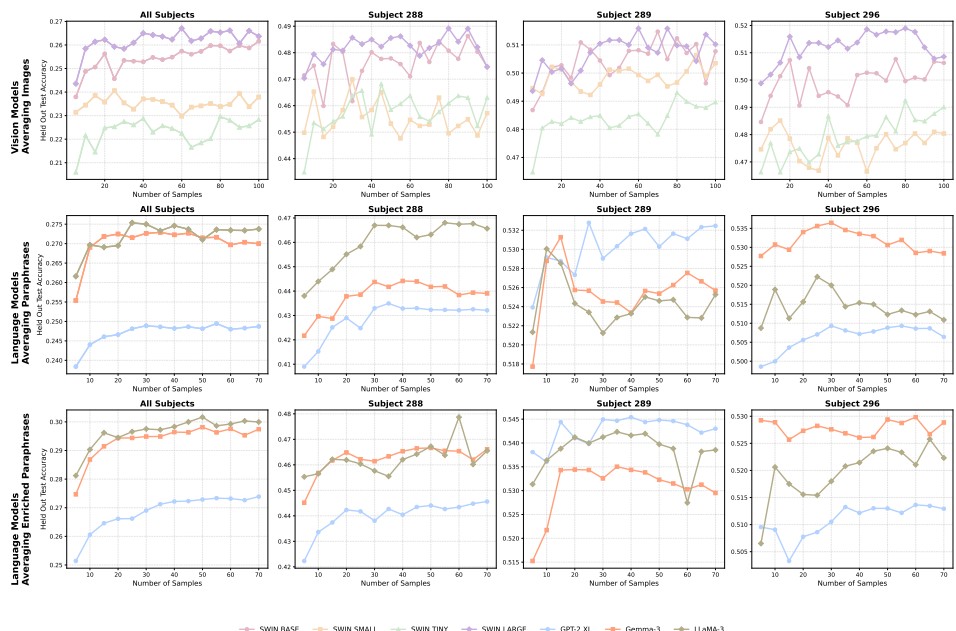

Figure 28: Pereira (2018) dataset - Trends observed when averaging more and more samples (images, paraphrases or enriched paraphrases) are consisted across individual subjects. We plot 3 out of 5 subjects of the Pereira dataset here.

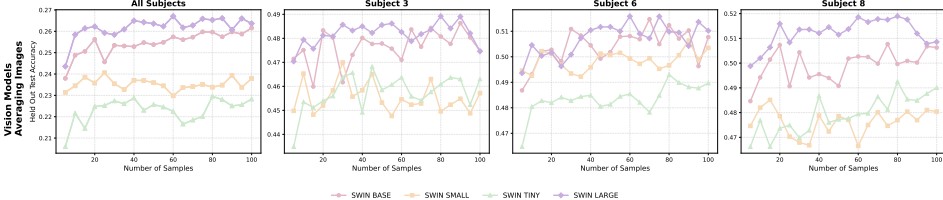

Figure 29: CSD (2025) dataset - Trends observed when averaging more and more samples (images) are consisted across individual subjects. We plot 3 out of 4 subjects of the CSD dataset here.

### A.7 INCREASING SAMPLES AND COMMONSENSE PARAPHRASES IMPROVES ENCODING MODEL ACCURACY: STATISTICAL EVIDENCE

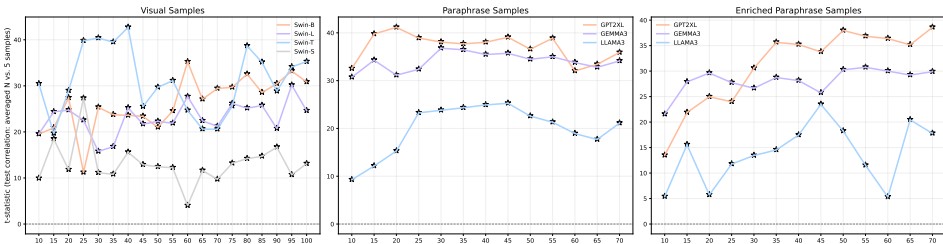

Figure 30: Statistical analysis showing that increasing the number of samples (images, paraphrases, or enriched paraphrases) improves encoding model performance. For each test split, we perform a paired t-test comparing the test accuracy scores of all voxels for encoding models trained with more than 5 samples ($N > 5$) versus models trained with 5 samples. The resulting t values are averaged across the three test splits, and the results are overall highly significant.

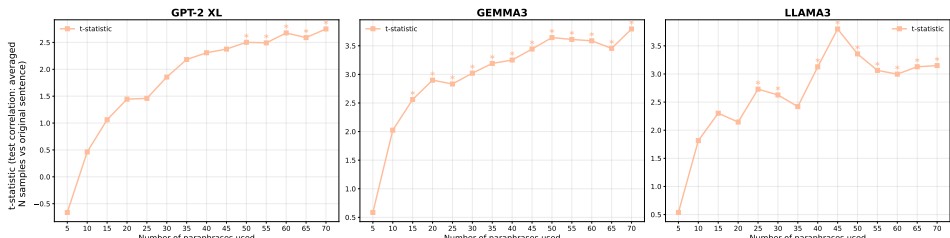

Figure 31: Expanded analysis of the experiments described in Section 3.2.2 across five data splits. For each condition, we compute test-time accuracies for averaged paraphrases and compare them with the accuracies obtained using the original single sentence. The t-statistics (significant onces marked with ∗) increase with the number of paraphrases, further supporting the claims in Section 3.2.2.

We additionally performed statistical analyses to support our central claim: increasing the number of samples, whether they are visual renderings of the original sentence, paraphrased versions of the sentence, or enriched paraphrases with added commonsense context—leads to a consistent improvement in encoding model performance. Although models trained on these transformed or cross-modal samples do not always match the performance of models trained directly on the original stimulus, they exhibit non-trivial predictive power, and their performance reliably increases as more samples are aggregated.

In Figure 30, we report a paired t-test comparing voxelwise held-out correlation scores from encoding models trained with more than five samples ($N > 5$) against those trained with only five samples. The ttest is performed using held out test scores across all voxels. The t and p values shown are averaged across all three test splits. We mark results with an asterisk when $p < 0.05$, indicating statistical significance. Across almost all conditions, the t values are positive and substantial, demonstrating that increasing the number of paraphrased or augmented samples has a measurable effect on model accuracy.

Notably, we do not observe a strictly monotonic increase in the t statistic as the number of samples grows. This pattern is expected, as adding more samples can sometimes introduce additional variability, so an increase of 15 samples might, in certain cases, result in a more significant change than adding 60 samples. However, the fact that the t values remain consistently positive indicates that, overall, increasing the sample size provides a clear advantage. Together, these statistical results reinforce our claim that expanding the sample set yields meaningful improvements in encoding model performance.

Further, we claim that adding commonsense paraphrases substantially improves the performance of encoding models beyond what is achieved by models trained on the original stimulus alone. To

support this claim, we repeated the experiments described in Section 3.2.2 using five data splits. For each split, we computed a held-out test accuracy score, resulting in two arrays of length 5: one for encoding models trained on enriched paraphrases and one for models trained on the original sentences. We then calculated the t statistic between these two arrays. We observed an overall increase in t values as more paraphrases were incorporated (Figure 31), particularly for larger models. These results further strengthen our claim that incorporating commonsense paraphrases benefits encoding model performance.

### A.8 Ablation Analyses Confirm that Performance Gains Reflect Meaningful Information

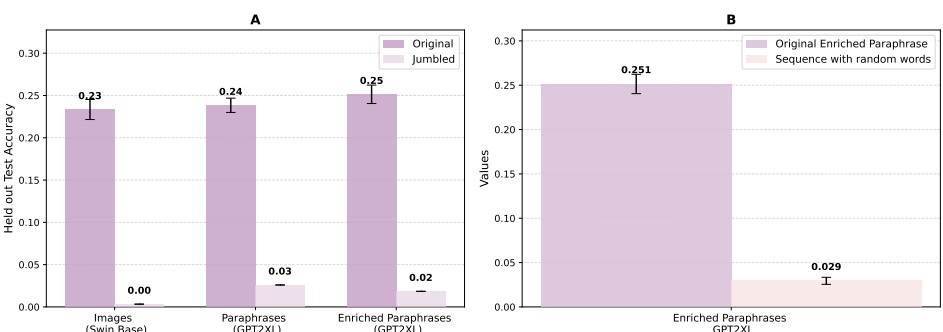

Figure 32: Pereira (2018) Dataset - (A) Comparison of encoding models trained on true input-output pairs versus randomly paired inputs and outputs. Models trained with true pairs substantially outperform those trained with random pairs, (B) Comparison of encoding models trained on original enriched paraphrases versus sequences of random words. Models trained with enriched paraphrases markedly outperform those trained with random word sequences.

To ensure that the performance gains observed with increasing the number of samples (visual samples, paraphrases, or enriched paraphrases) reflect genuinely meaningful information—rather than arbitrary improvements due to simply adding more data, we conducted a series of ablation analyses.

First, as a basic control, we disrupted the true stimulus–response correspondence by randomly shuffling stimulus–voxel pairs during training. As expected, training the encoding model on these mismatched pairs led to a dramatic drop in performance (Figure 32-A), confirming that the benefits of larger sample sets depend on the semantic relevance of the added information.

As noted earlier in the manuscript, the generated visual samples and paraphrases do not introduce any information beyond what is already contained in the original sentence input. In contrast, the enriched paraphrases were specifically designed to incorporate additional commonsense knowledge typically implied by the stimulus but not explicitly stated. These enriched descriptions are therefore longer and richer than the original sentences or their simple paraphrases. To test whether the observed performance improvements stem from this added semantic content, and not merely from increased sequence length—we performed three additional ablations:

1. **Length-matched random word sequences** Instead of enriched paraphrases, we generated sequences of random English words matched in length to the enriched paraphrases (five per input). As expected, encoding models trained on these random sequences performed extremely poorly—near chance (Figure 32-B), and far below models trained on enriched paraphrases.

2. **Original sentence + appended random words** To further disentangle the effects of additional information from those of increased sequence length, we constructed control sequences that appended random English words to the original sentence until their total length matched that of the enriched paraphrases. For each input, we generated 5, 50, and 70 such length-matched controls. As shown in Figure 33, these appended-word controls improved performance relative to the unmodified original sentence, however, they still performed substantially worse than the enriched paraphrases, with the discrepancy becoming even

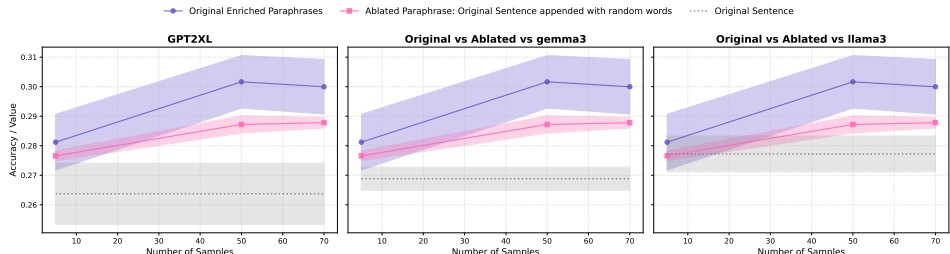

Figure 33: Pereira (2018) Dataset - Comparison between encoding models trained on (1) original sentence stimulus, (2) enriched paraphrases and (3) ablated paraphrases. The ablated paraphrases were constructed by appending random words to the original sentence to match the length of the enriched paraphrases. Models trained on enriched paraphrases outperform those trained on ablated paraphrases, highlighting that the performance gains stem from added semantic information rather than from an arbitrary increase in sentence length.

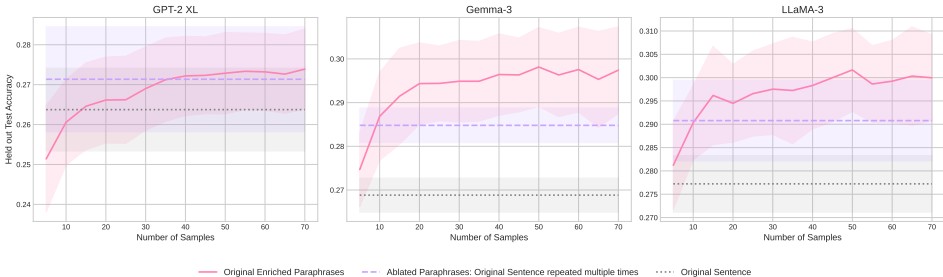

Figure 34: Pereira (2018) Dataset - Comparison between encoding models trained on (1) original sentence stimulus, (2) enriched paraphrases and (3) ablated paraphrases. The ablated paraphrases were constructed by repeated the original sentence multiple times to match the length of the enriched paraphrase. Models trained on enriched paraphrases outperform those trained on ablated paraphrases, highlighting that the performance gains stem from added semantic information rather than from redundant increase in sentence length.

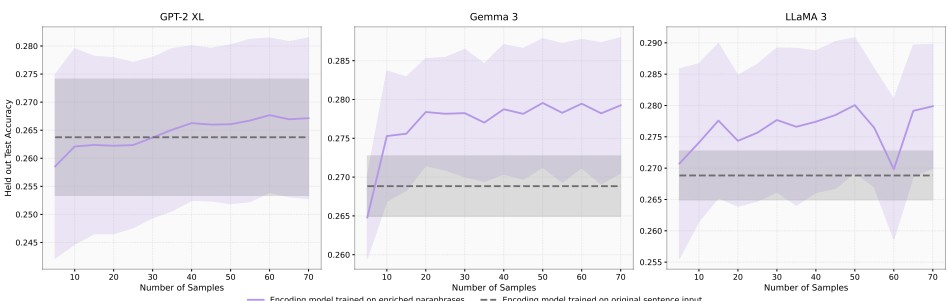

Figure 35: Pereira (2018) Dataset - Comparison of encoding models trained on original sentence stimulus with those trained using enriched paraphrases. LLM embeddings of the original sentence stimulus is used as input during inference. Encoding models trained with enriched paraphrase embeddings tend to improve in performance as more and more paraphrase samples are used during training.

more pronounced in larger models such as Gemma and Llama compared to GPT-2-XL. This demonstrates that the advantage of enriched paraphrasing cannot be explained by sequence length or arbitrary additional content alone.

3. **Length-matched repetitions of the original sentence** Finally, to rule out the possibility that enriched paraphrases improve performance simply by introducing redundant informa-

tion that increases sequence length, we created control sequences by repeating the original sentence until their total length matched that of the enriched paraphrases (five samples per input). Due to the limited number of unique repetitions possible, this ablation was restricted to five samples. As shown in Figure 34, these repeated-sentence controls improved performance relative to the original stimulus, however, performance remained substantially below that of the enriched paraphrases, particularly for larger models, confirming that redundancy alone cannot account for the observed gains.

These ablations collectively underscore the semantic richness of the enriched paraphrases, demonstrating that the observed performance improvements are driven by meaningful information rather than superficial factors such as increased sequence length or redundancy. To further validate this, we performed an additional analysis: encoding models trained on the enriched paraphrases were used to predict brain responses to the original sentences. Interestingly, these models substantially outperformed models trained solely on the original sentences (Figure 35), indicating that incorporating additional commonsense context allows the model to better capture the core semantic content of the original input. This finding highlights the value of enriched paraphrases not only in enhancing model performance on extended or augmented stimuli, but also in improving generalization to the original, unaltered stimuli by providing a richer, more structured representation of the underlying meaning.

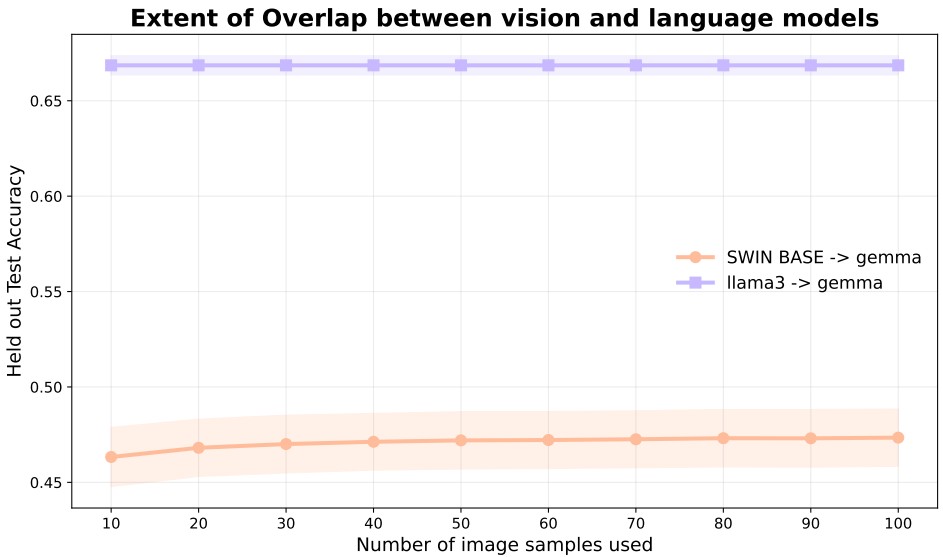

Figure 36: Pereira (2018) dataset: We compared how well representations from a vision model (Swin) versus a different language model (LLaMA) predict embeddings from another LLM (Gemma). Predicting Gemma using LLaMA yields substantially higher accuracy (0.66) than predicting Gemma using Swin even as we aggregate information across multiple visual exemplars (0.45). This gap demonstrates that, even within model space, vision and language models are far from interchangeable and do not share a single representational geometry.

## A.9 THE USAGE OF LARGE LANGUAGE MODELS (LLMS)

LLMs are primarily used to identify typos and make the language more aligned with conventions of academic writing.

