# OpenReview forum: "Modeling the language cortex with form-independent and enriched representations of sentence meaning reveals remarkable semantic abstractness"
_ICLR.cc/2026/Conference — Submitted to ICLR 2026_

### Official Review · Reviewer_Hsss · 2025-10-25

**Soundness:** 3
**Presentation:** 3
**Contribution:** 3
**Rating:** 4
**Confidence:** 4

**Summary:**

This paper investigates how abstract meaning is represented in the human language cortex by training ridge regression encoding models that predict fMRI responses from embeddings derived from large language models and vision models. The authors introduce a form-independent modeling framework that evaluates whether diverse inputs, such as paraphrases, enriched contextual sentences, and generated images, can predict neural activity in the language cortex. The study shows that averaging across multiple paraphrases or image embeddings enhances encoding accuracy and that semantically enriched sentences can even surpass the original text representations.

**Strengths:**

* The work investigates a fundamental question in cognitive neuroscience and AI — whether the brain encodes semantic meaning in a form-independent and modality-independent manner.

* Using representational augmentations across different forms and modalities (such as multiple paraphrases and generated images) provides a novel approach to probing semantic abstraction in brain representations.

**Weaknesses:**

* The correlation-based encoding models cannot establish whether the brain’s “abstract” representation is genuinely modality-independent or form-independent.

* The paper reports that representational diversity (such as enriching sentences) substantially improves encoding accuracy. How do the authors ensure that these gains arise from adding meaningful, semantically relevant information rather than simply increasing input length or redundancy? The paraphrases and images generated by large models may include noise or implausible details; quality control and human validation are insufficiently described.

* In several experiments, purely visual models (e.g., Swin Transformers) outperform CLIP in predicting language-cortex responses, even though CLIP is explicitly trained for vision–language alignment. Could the authors clarify why this occurs?

* The paper reports many comparisons and trends (e.g., improvement with more paraphrases or images) but lacks formal statistical testing of key effects.

* The paper is dense, and main findings could be summarized more clearly and concisely.

**Questions:**

* How consistent are the findings across individual subjects and brain regions (beyond aggregate “language cortex” analyses)?

* While the paper provides a comprehensive list of the visual models (such as ResNet, DINO, and Swin Transformer) in Appendix A.4, it does not specify the source or configuration of their pretrained checkpoints. For instance, it remains unclear whether the Swin Transformers were initialized from ImageNet-1K or ImageNet-22K.

* How were the enriched paraphrases validated to ensure they add plausible commonsense information rather than arbitrary details?

* Given the low temporal resolution of fMRI, how confident can we be that the observed improvements truly reflect more abstract or enriched semantic representations rather than temporal averaging effects or hemodynamic smoothing?

---

> ### Author Response · Authors · 2025-11-21
> **Official Response to Reviewer Hsss (1/4)**
>
> We thank the reviewer for their thoughtful comments and constructive suggestions to improve our paper. We have addressed each of the weaknesses and questions in detail below. In accordance with the rebuttal guidelines, we have completed as many additional analyses as possible by November 20th, and we will incorporate further experiments if time permits during the remainder of the review period.
>
> We apologize for the dense presentation in the original submission. To improve clarity and readability, we have expanded the introduction to include additional context and related work, including discussion of the Platonic Representation Hypothesis Huh et al., 2024 and Nikolaus et al. (2024). We have also substantially extended the Appendix with detailed descriptions of the datasets, training objectives, and model architectures **(Sections A.1–A.4)**. We also add detailed statistical tests **(Appendix section A.7)** and ablation studies **(Appendix section A.8)** to the various samples (paraphrases and images) encode semantically relevant information rather than noise, and that the improvement in modeling performance using commonsense enhanced paraphrases stems from enriched information content rather than from incidental increases in sentence length. Finally, we now reference concrete examples in the Methods section for each type of representation used to model the language cortex, to make it more intuitive. **All modifications to the main paper are highlighted in blue.**
>
> Detailed response to the review is as follows -
>
> ***The correlation-based encoding models cannot establish whether the brain’s “abstract” representation is genuinely modality-independent or form-independent. (Weakness 1) -***
>
> We appreciate the reviewer raising this important conceptual point and want to clarify what our results do and do not claim.
>
> What our encoding analyses reveal is that the predictive dimensions currently accessible to computational models do not require exact form preservation.  When we strip away surface form through cross-modal transformation (text→images), paraphrasing, or semantic enrichment, we retain substantial predictive power. Critically, we observe this across diverse vision and language models suggesting this reflects a property of high-level semantic structure rather than any specific architecture.
>
>
> Our goal is to characterize which representational spaces align with language cortex activity, not to make strong mechanistic claims about how the brain implements or uses these representations. Correlation-based encoding models are the standard approach for such questions throughout the neuroimaging literature (Huth et al. 2016; Schrimpf et al. 2020; Goldstein et al. 2022). Like those studies, we draw inferences about representational structure through systematic comparison: Which types of representations predict more variance? How does prediction change when we manipulate semantic content versus surface form? Our findings show that language cortex responses can be predicted from more abstract and diverse representational sources than previously demonstrated **(lines 475-478)**. The novel aspects are: (1) successful cross-modal prediction through vision models, (2) systematic improvement through representational averaging, and (3) enhanced prediction through semantic enrichment beyond what's explicit in text. Critically, our results advance understanding relative to prior work: by demonstrating that language cortex activity can be predicted not only from LLM embeddings but also from visual and paraphrased representations, we reveal that the semantic structure in language cortex is more abstract than previously characterized using linguistic models alone.
>
>
> Importantly, we are not asserting that form information is absent from language cortex responses. Rather, we are demonstrating that the way the language cortex represents meaning is abstract enough to be well-modeled by diverse representational sources, including high-level visual abstractions. Indeed, as we note in the Introduction, prior work consistently shows that original sentences yield higher responses than manipulated variants (word lists, scrambled syntax), highlighting that compositional structure matters for the language cortex **(lines 68-75)**. It remains possible that future models encoding form information more brain-like would lose significant predictivity when form is abstracted away or achieve much higher predictivity than current models.

---

> ### Author Response · Authors · 2025-11-21
> **Official Response to Reviewer Hsss (2/4)**
>
> *Continued from above:*
>
> We have revised the manuscript to more carefully frame our claims around "the degree of abstraction in model-predictable cortical variance" (see discussion highlighted in blue) and acknowledged that our correlation-based approach does not establish absolute form-independence in representations of meaning or address how the brain functionally implements or uses these representations. For example, we explicitly state in the discussion now **(lines 501-513)**.
>
> *‘Several methodological constraints should be acknowledged. First, our claims characterize the degree of abstraction in model-predictable cortical variance rather than making absolute assertions about form-independence in the language cortex's representations of meaning. The success of diverse representational sources (vision models, paraphrases, enriched content) demonstrates that the components of neural responses currently accessible to computational models reflect relatively abstract semantic structure. However, the residual advantage of original sentences indicates that form-specific information contributes to neural responses, even if less accessible to current embeddings. Our correlation-based encoding approach reveals which representational spaces align with language cortex activity but cannot establish whether the brain's meaning representations are genuinely independent of linguistic form, nor can it reveal how these representations support comprehension or behavior.’*
>
>
> ***Quality of generated samples (Weakness 2, Question 3) -***
>
> We thank the reviewer for raising these important methodological concerns. We have now added extensive ablation studies in Appendix A.8 to demonstrate that the performance gains observed with our generated samples (images, paraphrases and enriched paraphrases) arise from semantically meaningful information, rather than from random noise or superficial changes to the input (Figure 32). Specifically, we show that the improvements enabled by enriched paraphrases with added commonsense context stem from their information richness, not from simply increasing input length or redundancy.
>
> To validate this, we compare the performance of models trained on the actual enriched paraphrases with two ablated controls:
>
> 1. **Arbitrary length increase**: The original sentence is appended with random English words to match the length of the enriched paraphrases **(Figure 33)**.
> 2. **Redundant length increase**: The original sentence is repeated multiple times to match the paraphrase length **(Figure 34)**.
>
> Both length-matched controls show modest improvements over the original-sentence baseline. One plausible explanation is that longer inputs can subtly affect the model’s internal aggregation e.g., by altering attention patterns or distributing contextual weighting across more tokens, which can slightly stabilize the resulting embedding even when no new semantic content is added. Critically, however, enriched paraphrases substantially outperform both ablated variants in almost all cases. This clearly demonstrates that the observed gains stem from meaningful, semantically relevant information rather than from input length or incidental processing changes introduced by longer sequences.
>
> Interestingly, we further find that encoding models trained on enriched paraphrases generalize better to the original stimuli: when predicting brain responses to the original sentences, these models substantially outperform models trained solely on the original text **(Figure 35)**. This result underscores that incorporating additional commonsense context helps the model capture the core semantic content more effectively. The enriched paraphrases not only boost performance on the enriched stimuli but also enhance generalization to the original, unaltered sentences by providing a richer and more structured representation of meaning.
>
> For images, we used CLIP Score (cosine similarity between CLIP vision and text embeddings), a widely validated metric correlated with human judgments of image-caption alignment and a leading automated measure for text-to-image generation quality. Our analyses confirm CLIP Score reliably ranks sample quality: selecting images in descending order of CLIP Score yields significantly faster accuracy gains compared to random sampling **(Figures 4, 5, 21, 22)**, demonstrating systematic quality variation that directly impacts neural prediction.

---

> ### Author Response · Authors · 2025-11-21
> **Official Response to Reviewer Hsss (3/4)**
>
> *Continued from above*
>
> Finally, although all generated samples used in this paper were manually inspected by the authors, we acknowledge the importance of a more formal human evaluation. We are currently conducting a small-scale survey in which, for each scenario, we uniformly sample five outputs across the range of CLIP scores and ask human raters to assess their relevance to the original sentence. We will then correlate these ratings with the corresponding CLIP scores to further validate the semantic fidelity and overall quality of the generated samples. We aim to complete this analysis before the end of the rebuttal period, time permitting; otherwise, we will include the full evaluation in the camera-ready version should the paper be accepted.
>
>
> ***Why purely visual models (e.g., Swin Transformers) outperform CLIP in predicting language-cortex responses, even though CLIP is explicitly trained for vision–language alignment?  (Weakness 3) -***
>
> This finding was initially surprising to us as well, but we believe it reflects fundamental differences in what these models optimize for and how those objectives align with representations in the language cortex. A plausible explanation for why unimodal vision models (e.g., Swin) outperform CLIP is that both the language system and modern vision transformers learn high-level conceptual structure from rich co-occurrence statistics in their respective domains. Swin models, trained solely on large-scale visual datasets, may preserve more of this broad conceptual variability than CLIP, whose representations are explicitly optimized for image–text alignment. A further clue is that CLIP’s own text encoder is among the weakest language models in predicting language-cortex responses. This suggests that CLIP’s contrastive objective designed to match images to captions compresses or discards linguistic dimensions that matter most for the brain. We therefore view this pattern not as a weakness but as an intriguing result: conceptual abstractions learned from vision alone can approximate aspects of linguistic meaning, motivating future work comparing how different training objectives shape convergence with brain representations.
>
>
> To acknowledge this pattern more clearly, we have added a brief note in the Results section highlighting that unimodal vision models outperform CLIP in predicting language-cortex responses, and that this likely reflects differences in their training objectives rather than a methodological artifact **(lines 214-218 and 251-258)**.
>
> ***Statistical Testing of Key effects (Weakness 4) -***
>
> We have now added formal statistical tests (Appendix A.7) to support the key effects highlighted in the paper: increasing the number of samples, whether they are visual renderings of the original sentence, paraphrased versions of the sentence, or enriched paraphrases with added commonsense context, leads to a consistent improvement in encoding model performance. Although models trained on these transformed or cross-modal samples do not always match the performance of models trained directly on the original stimulus, they exhibit non-trivial predictive power, and their performance reliably increases as more samples are aggregated.
>
> In **Figure 30,** we report paired t-tests comparing voxelwise held-out correlation scores from encoding models trained with more than five samples (N > 5) against models trained with only five samples, using held-out test scores across all voxels (averaged over all three test splits). Conditions with p < 0.05 are marked with an asterisk. Across nearly all settings, t-values are highly significant, indicating that aggregating more visual renderings, paraphrases, or enriched paraphrases reliably improves encoding performance.
>
> Notably, we do not expect the t-statistic to increase monotonically with sample count, as performance often plateaus once sufficient semantic signal is captured, and additional samples can introduce minor variability. What matters for our central claim is that the effect is consistently positive and statistically significant across models and stimulus types. Together, these statistical results reinforce our claim that expanding the sample set yields meaningful improvements in encoding model performance.

---

> ### Author Response · Authors · 2025-11-21
> **Official Response to Reviewer Hsss (4/4)**
>
> Further, we claim that adding commonsense paraphrases substantially improves the performance of encoding models beyond what is achieved by models trained on the original stimulus alone. To statistically test this effect, we repeated the paraphrase experiments **(Section 3.3.2)** across five data splits, yielding five held-out accuracy scores for models trained on enriched paraphrases and five for models trained on the original sentences. We then computed paired t-tests between these two arrays. As shown in **Figure 31**, t-values increase as more enriched paraphrases are incorporated, with the strongest effects for larger models. These results provide convergent statistical evidence that adding commonsense context yields a reliable improvement in encoding performance beyond using the original sentence alone.
>
>
> ***Weakness 5 - How consistent are the findings across individual subjects and brain regions (Question 1)?***
>
> Our findings are quite consistent across individual subjects, and we have added examples for the same across Pereira dataset **(Figure 28)** and the CSD dataset **(Figure 29)** with a small subsection of the total models used.
>
> We also appreciate the suggestion to examine finer-grained regional differences. However, prior work indicates that such dissociations within the core language network are minimal. Across thousands of sentences, Tuckute et al. (2024; Fig. 4b,c) show that responses in the regions we analyze, including the Inferior Frontal Gyrus (pars opercularis and orbitalis), Middle Frontal Gyrus, Anterior Temporal Cortex, and Posterior Temporal Cortex, are highly correlated, exhibiting near-parallel response profiles. This pattern is reinforced by a large body of evidence showing that virtually all core language regions respond similarly to linguistic manipulations, both in controlled experiments (Rodd et al., 2010; Fedorenko et al., 2020; Blank & Fedorenko, 2020; Hu et al., 2022) and during naturalistic comprehension (Blank et al., 2016; Shain et al., 2020; Wehbe et al., 2021; Shain et al., 2022). Aside from the angular gyrus which is often considered outside the core language network, there is little evidence for functional dissociations within the network.
>
> Given this well-established functional homogeneity, averaging predictivity across the language network provides a reliable and interpretable summary measure. A comprehensive region-by-region analysis would therefore be unlikely to alter the main conclusions, and conducting it is beyond the scope of the 9-page format. We plan to include such analyses in an extended version of the paper.
>
> ***Question 2*** - We have now updated **Appendix A.4** with the source and configuration of the pretrained checkpoints.
>
> ***Given the low temporal resolution of fMRI, how confident can we be that the observed improvements truly reflect more abstract or enriched semantic representations rather than temporal averaging effects or hemodynamic smoothing? (Question 4)***
>
> While fMRI’s low temporal resolution arising from temporal averaging and hemodynamic smoothing is a known limitation, these factors are not unique to our study; they are standard across essentially all work on the language network. Seminal studies (Huth et al., 2016) demonstrating semantic selectivity in this system likewise rely on temporally averaged BOLD responses to isolated sentences. Importantly, our static sentence-presentation paradigm avoids many confounds present in naturalistic continuous-speech experiments (e.g., stories or audiobooks), where long-range contextual dynamics, discourse structure, and memory processes introduce complexities that cannot be cleanly controlled or modeled within our framework. We acknowledge that extending these analyses to modalities with higher temporal precision (e.g., ECoG) would be valuable future work; however, most available public ECoG datasets (e.g. Zada et al., 2025) involve narrative stimuli, which come with the same uncontrolled contextual factors noted above.
>
>
> We now explicitly discuss the temporal-resolution limitations of fMRI in the revised Discussion section **(highlighted in blue, lines 529-533)**.

---

### Official Review · Reviewer_tWEL · 2025-10-30

**Soundness:** 3
**Presentation:** 2
**Contribution:** 2
**Rating:** 4
**Confidence:** 4

**Summary:**

This paper shows that fMRI response to sentence stimuli in the brain’s language network can be modeled by not only LLM embeddings of the original sentences, but also by vision model embeddings of images generated based on sentence prompts, and LLM embeddings of paraphrased sentence. The predictivity of language and vision models is also comparable when considering only content or header words in the original sentence. Finally, they find that brain predictivity can be improved when considering “enriched paraphrases” beyond the original sentence.

**Strengths:**

The paper uses state-of-the-art AI models to answer important questions about the human language network

The paper compares a wide range of multimodal model inputs to compare different sources of semantic information

The “enriched paraphrase” results are interesting and novel and suggest ways the language network extracts meaning beyond the linguistic input.

**Weaknesses:**

A growing body of work has now shown that large language and vision models learn overlapping representations of iamges and their captions (e.g., Platonic Representation Hypothesis Huh et al 2024). This has been leveraged in similar cognitive neuroscience studies to show that LM embeddings of sentence captions can predict the brain’s visual cortex responses (Doerig et al., 2025, Conwell et al. 2025). The authors do not cite this large body of relevant prior work. They also do not report the correlation / overlap between LM and VM embeddings in any of hte experiments, but given that LM and VM model representations are likely overlapping, the bulk of the results in this paper are not particularly surprising. Much of the discussion (and title) describing these as “independent” are incorrect/misleading. For each experiment, the authors should quantify the overlap between the “independent” sources of information and adjust their intepretation accordingly.

While based on prior work, the authors should provide more context on the fMRI experiments so the paper is more easily interpretable / can be read on its own. In particular, details/differences between the different datasets and the discussion of “header words” in the Pereira dataset were hard to follow.

Throughout, the authors show only mean predictivity across the entire language network, a large swath of cortex averaging over hundreds (or thousands?) of voxels. It would help to analyze the data in a more fine-grained way to see if there are particular regions that are sensitive to different types of information.

The figures throughout could be made more readable. Eg the CLIP dots/lines are very difficult to see in most figures.

The paper could use some more technical details / motivation for the different model types. For example, a brief overview of the different model architectures and why they are included (just for variety?)

**Questions:**

As detailed above, the most important question to address in each experiment is the extent of overlap between the vision and language model embeddings, without this it is impossible to assess the novelty/significance of the paper.
How was image quality assessed/quantified in figure 4?
The results of the “enriched paraphrase” are very interesting, but how does this reconcile with the large body of prior work suggesting that the human language network does not represent extralinguistic knowledge?

---

> ### Author Response · Authors · 2025-11-21
> **Official Comment to Reviewer tWEL (1/2)**
>
> We thank the reviewer for their thoughtful comments and constructive suggestions to improve our paper. We have addressed each of the weaknesses and questions in detail below. In accordance with the rebuttal guidelines, we have completed as many additional analyses as possible by November 20th, and we will incorporate further experiments if time permits during the remainder of the review period.
>
>
> We apologize for the dense presentation in the original submission. To improve clarity and readability, we have expanded the introduction to include additional context and related work, including discussion of the Platonic Representation Hypothesis Huh et al., 2024 and Nikolaus et al. (2024). We have also substantially extended the Appendix with detailed descriptions of the datasets, training objectives, and model architectures **(Sections A.1–A.4)**. We also add detailed statistical tests **(Appendix section A.7)** and ablation studies **(Appendix section A.8)** to the various samples (paraphrases and images) encode semantically relevant information rather than noise, and that the improvement in modeling performance using commonsense enhanced paraphrases stems from enriched information content rather than from incidental increases in sentence length. Finally, we now reference concrete examples in the Methods section for each type of representation used to model the language cortex. All modifications to the main paper are highlighted in blue. We have updated **Figures 2,6,7** with darker shades so that each of the model performances can be seen more clearly.
>
> Detailed response to the review is as follows -
>
> ***How was image quality assessed/quantified in figure 4?*** The image quality in **Figure 4** is quantified using the cosine similarity between the CLIP vision embedding of each image and the CLIP text embedding of its corresponding caption (a metric widely known as CLIP Score). This procedure is described in the **caption of Figure 4** and in the main text **(lines 261–262)**. CLIP Score has been validated against human judgments of image–caption similarity and is one of the leading automated metrics for assessing semantic fidelity in text-to-image generation. For additional discussion regarding the quality of the generated samples, please refer to our response to **Reviewer Hsss**.
>
>
> ***Regarding overlapping representations between images and their captions -***
>
> We thank the reviewer for raising this point. We now cite the relevant literature on representational convergence between vision and language models, as well as studies showing that linguistic captions can predict responses in high-level visual cortex in the introduction **(highlighted in blue, lines 76-88)**.
>
>
> That said, we emphasize that the fact that language models encode meaning in a way that partially resembles unimodal vision models does not imply that the human language cortex will do the same. Representational similarity is not transitive: overlap between VM–LM and overlap between LM–language-cortex does not guarantee overlap between VM–language-cortex. The dimensions shared by vision and language models may reflect high-level semantic structure, whereas the dimensions shared by language models and the language cortex may reflect syntactic, relational, or compositional information that vision models never acquire. In short, these two alignments can depend on entirely different subspaces.
>
>
> Thus, similarity between LMs and VMs places no constraint on whether the language cortex should share that structure. The language system could, in principle, represent meaning in a format that is more abstract, more relational, more syntactic, or more modality-specific than anything present in either model. Each pair of systems (VM–LM, LM–language-cortex) can align on different components, just as two variables might both correlate with temperature, and one might correlate with humidity, without implying that the other correlates with humidity.
>
>
> We also conducted the additional analysis suggested by the reviewer **(Appendix Figure 36)**: we compared how well representations from a vision model (Swin Base) versus a different language model (LLaMA) predict embeddings from another LLM (Gemma). Predicting Gemma using LLaMA yields substantially higher accuracy (0.66) than predicting Gemma using Swin even as we aggregate information across multiple visual exemplars (0.45). This gap demonstrates that, even within model space, vision and language models are far from interchangeable and do not share a single representational geometry. This makes it all the more striking that unimodal vision models nonetheless approach large LMs in predicting language-cortex responses. This is not something that follows from representational convergence in model space; rather, it speaks to the structure of meaning encoded in the human language system itself.

---

> > ### Comment · Reviewer_tWEL · 2025-11-21
> >
> > Thank you for these revisions and responses to my review. I believe the added discussion about overlap between VM-LM representations helps emphasize and contextualize the contributions. I have updated my score.

---

> ### Author Response · Authors · 2025-11-21
> **Official Comment to Reviewer tWEL (2/2)**
>
> *Continued from above*
>
> Finally, although prior studies have shown that LLM embeddings can predict responses in visual cortex, to our knowledge no prior work has demonstrated the reverse: that unimodal vision models can capture activity in the language network. Our findings directly establish this nontrivial result.
>
> ***Throughout, the authors show only mean predictivity across the entire language network, a large swath of cortex averaging over hundreds (or thousands?) of voxels. It would help to analyze the data in a more fine-grained way to see if there are particular regions that are sensitive to different types of information.***
>
>
> We appreciate the suggestion to examine finer-grained regional differences. However, prior work indicates that such dissociations within the core language network are minimal. Across thousands of sentences, Tuckute et al. (2024; Fig. 4b,c) show that responses in the regions we analyze, including the Inferior Frontal Gyrus (pars opercularis and orbitalis), Middle Frontal Gyrus, Anterior Temporal Cortex, and Posterior Temporal Cortex, are highly correlated, exhibiting near-parallel response profiles. This pattern is reinforced by a large body of evidence showing that virtually all core language regions respond similarly to linguistic manipulations, both in controlled experiments (Rodd et al., 2010; Fedorenko et al., 2020; Blank & Fedorenko, 2020; Hu et al., 2022) and during naturalistic comprehension (Blank et al., 2016; Shain et al., 2020; Wehbe et al., 2021; Shain et al., 2022). Aside from the angular gyrus which is often considered outside the core language network, there is little evidence for functional dissociations within the network.
>
>
> Given this well-established functional homogeneity, averaging predictivity across the language network provides a reliable and interpretable summary measure. A comprehensive region-by-region analysis would therefore be unlikely to alter the main conclusions, and conducting it is beyond the scope of the 9-page format. We plan to include such analyses in an extended version of the paper.
>
>
> ***The results of the “enriched paraphrase” are very interesting, but how does this reconcile with the large body of prior work suggesting that the human language network does not represent extralinguistic knowledge?***
>
> This is an important question. At first glance, it may seem puzzling: if the language network represents only the linguistic content of the sentence (and not rich, extralinguistic world knowledge), why does giving the model extra commonsense detail - detail the participant never saw - improve prediction of language-evoked activity? This puzzle arises only under a very sharp dichotomy: either (i) the language cortex is purely shallow or string-bound, or (ii) if enrichment helps, the system must encode full-blown encyclopedic knowledge. We believe neither extreme is accurate. There is broad agreement that the language network does not represent arbitrary extralinguistic facts or detailed visual knowledge, but this does not mean it encodes only surface details.
>
>
> During comprehension, the system builds structured interpretations of sentences - who did what to whom, what event is being described, and whether the scenario is typical or plausible. These are linguistically driven representations, but they inevitably draw on broad statistical regularities learned through language exposure. In other words, the language network constructs event-level meaning from the sentence, not just a literal, minimal reading of the words.
>
> Our enriched-paraphrase results fit naturally within this middle ground. The brain responses we model are always to the original sentence; enrichment only changes the model’s embedding. Adding stereotypical contextual detail pushes the model’s representation toward the prototypical event structure that people implicitly recover when interpreting the original linguistic input. This allows the model to better approximate the brain data without implying that the language cortex explicitly encodes all the added information as standalone conceptual knowledge.
>
> To address the reviewer’s concern, we have made minor clarifications in the Discussion (highlighted in blue) noting that enriched paraphrases act as a tool for steering model embeddings toward the latent event structure recovered during comprehension, rather than suggesting that the language network represents arbitrary extralinguistic facts or encyclopedic knowledge **(lines 493-497).**

---

### Official Review · Reviewer_DsH1 · 2025-10-31

**Soundness:** 2
**Presentation:** 1
**Contribution:** 2
**Rating:** 2
**Confidence:** 5

**Summary:**

This manuscript asks if fMRI activity produced by sentences, as measured in the language network (here, several frontal and temporal regions) can be encoded from representations that differ from the original linguistic input. Specifically, they evaluate ability to predict language driven activity from embeddings from vision-only models (for matching pictures) and text models (operating over paraphrases and enriched text). The authors show that embeddings from vision models, generated from sentence-derived images, can predict sentence-evoked fMRI in language-selective cortex. They argue this demonstrates "remarkable abstractness" (see title) in the language system.

**Strengths:**

* The encoding pipeline is solid from an ML perspective
* The finding that averaging of sentence paraphrases improves predictivity is interesting (but see below)
* Contributes to an existing literature on abstract encoding of semantics in the brain

**Weaknesses:**

There is overstated novelty regarding documenting/discovering "abstractness" in language encoding . The title (“**remarkable abstractness**”) and Discussion (“**highly abstract**”) frame the main contribution as discovering abstract, cross-modal semantics. However, semantic abstractness across modalities has already been demonstrated, most directly in decoding studies showing cross-modal identification: Nikolaus et al. (not cited; https://arxiv.org/abs/2403.11771): show cross-modal fMRI-to-feature decoding between images and captions also when using unimodal vision features. Other, less recent cross-modal decoding results (all also not cited): Simanova et al. 2014; Shinkareva et al. 2011; Fairhall & Caramazza 2013 all show that category information generalizes over words/pictures/sounds in multiple regions. Importantly, many of those regions are also outside the language network as defined here.

I therefore argue that the central claim of discovering abstractness is not novel; what is novel is the encoding approach for sentence-level activity.

2. There is therefore a scholarship gap: the above literature is central to the paper’s framing, constrains the predictions, can guide the brain areas to be analyzed, but is currently missing. The manuscript should cite and relate to these studies. While one might argue that the focus on the language network as encoding abstractness is a contribution or focus of the paper, this is only valid if prior work is cited as prior related work, and furthermore, the prior work importantly suggests that in order to understand how the brain codes for modality-general information we would do well to look outside the language network in the first place.

3. I think there is an over-interpretation of the results relating to the effects of averaging paraphrases of the target sentence. The paper treats accuracy gains from averaging sentence paraphrases as evidence for "meaning amplification". A more parsimonious account is, instead, that it indicates variance reduction in the model space: paraphrase-specific idiosyncrasies (lexico-syntactic/style factors) would cancel under averaging, improving reliability of the embeddings rather than enriching semantics. As written, the manuscript does not try to distinguish enrichment from denoising. This can be done, for example, by adding the variance of paraphrases as an additional predictor.  If enrichment drives the result, increased variance should improve prediction, beyond the mean.

4. Re' methods: there is a potential circularity introduced by multimodal-trained encoders. Some tested models (nominally referred to as vision or text models) are trained with multimodal objectives (e.g., CLIP), which already align image–text embeddings. Using them to argue for brain abstractness is circular; the authors should use only unimodal encoders.

**Questions:**

none at this point

---

> ### Author Response · Authors · 2025-11-21
> **Official Response to Reviewer DsH1 (1/2)**
>
> We thank the reviewer for their thoughtful comments and constructive suggestions to improve our paper. We have addressed each of the weaknesses and questions in detail below. In accordance with the rebuttal guidelines, we have completed as many additional analyses as possible by November 20th, and we will incorporate further experiments if time permits during the remainder of the review period.
>
>
> To improve clarity and readability, we have expanded the introduction to include additional context and related work, including discussion of the Platonic Representation Hypothesis Huh et al., 2024 and Nikolaus et al. (2024). We have also substantially extended the Appendix with detailed descriptions of the datasets, training objectives, and model architectures **(Sections A.1–A.4)**. We also add detailed statistical tests **(Appendix section A.7)** and ablation studies **(Appendix section A.8)** to the various samples (paraphrases and images) encode semantically relevant information rather than noise, and that the improvement in modeling performance using commonsense enhanced paraphrases stems from enriched information content rather than from incidental increases in sentence length. Finally, we now reference concrete examples in the Methods section for each type of representation used to model the language cortex. **All modifications to the main paper are highlighted in blue.**
>
>
> Detailed response to the review is as follows -
>
> ***Weakness 1, 2 -***
>
> First, we want to clarify that our work does not ask how the brain codes for modality-general or cross-modal semantic information, nor do we test whether the language system responds similarly to visual depictions. This is not the question our study addresses.
>
> With that distinction in place, we note that Nikolaus et al. (2024) and related studies address a fundamentally different question from ours. They investigate where in the brain semantic information is accessible in a modality-general way i.e, which regions support cross-modal decoding when participants view images and read corresponding captions.  This is a question about cross-modal semantic localization. Decoding studies demonstrate that certain regions can support cross-modal identification, but they cannot reveal how much of the variance in language-evoked neural responses is explained by different representational hypotheses (original form, visual abstractions, paraphrases, enriched content). We also note that Nikolaus et al.’s own ROI results are consistent with a strongly modality-selective language system. In their language ROI, decoders recover captions well but perform poorly on images (image decoding accuracy is lowest there across models and decoder types), whereas robust modality-agnostic decoding is found only in high-level visual cortex. Thus, Nikolaus et al. do not show that the language system itself is modality-agnostic; rather, they localize cross-modal semantics primarily to high-level visual areas.
>
> In contrast, our work investigates the representational format of meaning within the language-selective system itself. We ask how linguistic semantics are encoded when the input is purely verbal, and whether this internal semantic structure is sufficiently abstract that it can be approximated by high-level visual conceptual representations. Crucially, we do not claim that the language system would respond similarly to visual depictions; rather, we probe whether the neural code evoked by text alone has an abstract structure that resembles high-level visual abstractions. These two questions - cross-modal semantic access versus the intrinsic representational format of linguistic meaning - are scientifically distinct and not addressed by the same analyses: success on one does not imply or constrain answers to the other.
>
> We thank the reviewer for raising this point; it helped us clarify the distinction between these questions more explicitly in the manuscript **(see the revised discussion section of the paper with changes highlighted in blue lines 501-514)**.
>
> We have also revised our framing in the abstract and introduction (highlighted in blue) to emphasize the novelty of the methodological pipeline we introduce **(lines 26-29 and lines 95-98)**: using diverse representational sources including unimodal vision models and enriched paraphrastic embeddings to model activity in the language-selective cortex at the sentence level. To our knowledge, no prior work has tested whether such representations can predict responses in the language network or compared their predictive power against standard language-model baselines.

---

> ### Author Response · Authors · 2025-11-21
> **Official Response to Reviewer DsH1 (2/2)**
>
> *Continued from above*
>
> ***Weakness 3 -***
>
> We believe the reviewer has misunderstood the central finding. Our claim is not that averaging paraphrases adds semantic enrichment. In fact, we explicitly acknowledge in the manuscript that most paraphrases differ from the original only in surface form and that averaging reduces surface-level idiosyncrasies. The key result is not that paraphrases contain new information - it is that even paraphrases that lack the original linguistic form, and that differ in syntax and lexical choice, nonetheless converge onto a representation that predicts neural responses nearly as well as the original stimulus.
>
> At minimum, our results show that the dimensions of LLM embeddings that predict activity in the language cortex do not rely heavily on the exact linguistic form. Even when lexical, syntactic, and phrasal structure are replaced by diverse paraphrastic alternatives, averaging these paraphrases recovers almost the same level of prediction accuracy as the original sentence. This cannot be explained by variance reduction alone: denoising paraphrastic noise cannot recreate the specific lexical items, syntactic frames, or phrasal structure of the original sentence. If those form-specific properties were essential for predicting neural responses, removing them would necessarily degrade performance rather than restore it through averaging.
>
> Building on this minimal observation, the most natural interpretation is that the aspect of linguistic meaning that drives responses in the language cortex is highly form-invariant. Diverse paraphrases converge onto a common semantic representation that remains aligned with the neural responses, suggesting that the meaning encoded by the cortex and captured by LLM-derived predictors is robust to substantial variation in surface form.
>
> Regarding the reviewer’s suggestion of including paraphrase variance as an additional predictor: we appreciate the idea, but we do not believe variance is an appropriate or interpretable neural predictor in this setting. The variance across paraphrases primarily reflects dispersion in the model’s embedding space, which is not a feature we expect the brain to encode or track.  In contrast, the mean embedding isolates the semantic information consistent across paraphrases and provides a more coherent representational hypothesis for the language cortex. Using variance as an additional predictor would therefore mix differences in sampling-induced spread with the actual semantic signal and is unlikely to yield interpretable or biologically meaningful improvements in prediction.
>
> Finally, we recognize that one sentence in the Discussion stating that averaging **“amplifies shared meaning while reducing noise from surface-level variations”** may have contributed to this misunderstanding. We have revised this phrasing to avoid implying semantic enrichment and to clearly convey that averaging primarily removes surface-level variability while isolating the semantic component.
>
> ***Weakness 4 -***
>
>
> We emphasize that our central claims are supported exclusively by unimodal vision encoders, not by CLIP or other multimodal models. CLIP was included only as a control to quantify how much explicit image–text alignment improves prediction. We found that in fact, it buys very little - CLIP vision encoder sometimes even underperforms compared to some vision models (Swin Transformers). We have clarified this in our methods section now: ‘We use CLIP strictly as a baseline to assess how much explicit image–text supervision improves prediction. Our core scientific conclusions are derived entirely from unimodal vision models, not from CLIP.’  **lines 214-218 and 251-258**

---

### Official Review · Reviewer_QaNP · 2025-10-31

**Soundness:** 4
**Presentation:** 4
**Contribution:** 4
**Rating:** 10
**Confidence:** 3

**Summary:**

The paper uses internal representations extracted from LLMs and well as from visual models to reconstruct activation patterns (fMRI) in the language-related brain regions of participants reading English/Chinese sentences. The comprehensive set of experiments investigate the impact of the using the representations based on the original texts, paraphrases thereof, images generated from the texts, as well as elaborations of the texts with additional, inferreble details. The findings suggest that the brain represents linguistic data in a largely modality independent way, while at the same time incorporating aspects of semantics which are not explicitly present in the input.

**Strengths:**

- The paper builds on a growing body of work using deep learning models to investigate the human brain. It uses this toolkit to address the question of how modality dependent and abstract the brain's representations are, in comprehensive detail.
- The methodology seems solid.
- The paper is written with admirable clarity and very easy to follow.
- The results are of broad interest to cognitive science.

**Weaknesses:**

I didn't indentify any substantive weaknesses. Some details of the experimental setup were missing, but can be easily added in a revision (see questions).

**Questions:**

- What exactly do you mean by abstract? Would a more precise terms be modality-independent or form-independent, as appropriate?
- How are sentence-level embeddings produced for the different models? For LLMs, does this involve aggretation token-level embeddings?
- Why were the embeddings extracted from the penultimate layer (in most or all cases as far as I can see). Was this based on a preliminary experiment?
- Lines 698-701: Can you elaborate on how combining two losses and using backprop for training is related to the presence of NaN values in the data?

---

> ### Author Response · Authors · 2025-11-25
> **Official Response to Reviewer QaNP**
>
> We are grateful to the reviewer for their encouraging feedback and thoughtful comments. Thank you!
> We have addressed each of the questions in detail below. In accordance with the rebuttal guidelines, we have completed as many additional analyses as possible by November 20th, and we will incorporate further experiments if time permits during the remainder of the review period.
>
>
> To improve clarity and readability, we have expanded the introduction to include additional context and related work, including discussion of the Platonic Representation Hypothesis Huh et al., 2024 and Nikolaus et al. (2024). We have also substantially extended the Appendix with detailed descriptions of the datasets, training objectives, and model architectures (Sections A.1–A.4). We also add detailed statistical tests (Appendix section A.7) and ablation studies (Appendix section A.8) to the various samples (paraphrases and images) encode semantically relevant information rather than noise, and that the improvement in modeling performance using commonsense enhanced paraphrases stems from enriched information content rather than from incidental increases in sentence length. Finally, we now reference concrete examples in the Methods section for each type of representation used to model the language cortex. All modifications to the main paper are highlighted in blue.
>
>
> ***What exactly do you mean by abstract? Would a more precise term be modality-independent or form-independent, as appropriate***
>
>
> By “abstract representations,” we refer to representations of meaning that are invariant to the specific form or modality of the input. Our results show that the dimensions of model embeddings that best predict language-cortex responses are those that remain stable across substantial variation in form (e.g., across diverse paraphrases) and can be approximated by high-level conceptual structure from nonlinguistic models. We are not claiming full modality-independence or form-independence. Rather, we use “abstract’’ to refer to a meaning representation that captures higher-level semantic structure beyond the exact lexical or syntactic realization of the sentence.
>
>
> ***How are sentence-level embeddings produced for the different models?*** For LLMs, does this involve aggregation token-level embeddings? These involve aggregated token level embeddings, and are further clarified in Appendix Section A.2
>
>
> ***Why were the embeddings extracted from the penultimate layer (in most or all cases as far as I can see). Was this based on a preliminary experiment?***
>
> Yes, we used embeddings extracted from the penultimate layer for all experiments reported in the paper, and this choice was informed by an initial pilot experiment that is not currently included in the manuscript. We will incorporate these details into the rebuttal if time permits, and we will ensure that they are included in the camera-ready version should the paper be accepted.
>
> ***Lines 698-701: Can you elaborate on how combining two losses and using backprop for training is related to the presence of NaN values in the data?***
>
> Using standard libraries to train a ridge regression encoder requires that the training data contain no NaN values. However, in the Pereira dataset, some voxel responses are NaN for specific stimuli but not others. In an N *M response matrix (with N stimuli and M voxels), removing an entire voxel/column would discard substantial data, even though only a small subset of entries is missing. This makes the closed-form ridge regression solution infeasible, since it cannot handle partially missing voxel values.
> For this reason, we adopt the same strategy used in prior work (Saha et al., 2025; Khosla & Wehbe et al., 2022) and train the encoding model using backpropagation instead of a closed-form solver. Our model is optimized using two losses, a mean squared error loss between predicted and observed voxel responses, and a correlation-based loss that encourages alignment between predicted and true voxel patterns. Crucially, both losses are applied in a masked manner: for any stimulus–voxel pair where the voxel response is NaN, the corresponding loss term is simply ignored. This allows us to fully utilize all available data without discarding voxels or stimuli, while still training a stable and well-behaved encoding model.

---

### Meta-Review · Area_Chair_XaDS · 2026-01-05

**Summary:**

This paper studies how the human language network represents abstract sentences and their semantics. This work uses LLMs and vision models to extract internal representations and reconstruct fMRI responses during sentence reading. The reviewers' main concerns were:
(i) framing and novelty: especially on the work’s overstating “remarkable semantic abstractness” given related work.
(ii) validation of the performance gains: whether the gains could be explained by more metrics rather than semantically meaningful additions.
(iii) methodological completeness and more ablations.
(iv) readability and clarity.
During rebuttal, the authors’ responses have improved the empirical results and reduced the concerns of (ii)-(iv), but an addressed concern remains around (i): even if the question differs from prior cross-modal decoding work, the reviewer(s) may still view the paper’s core claim as overstated. I suggest that this manuscript needs further revision, especially the statements on its novelty and framing, to achieve publication-ready quality.

**Reviewer Concerns:**

Concerns addressed in the rebuttal.

1.	Further clarification on what "abstract" means and details on embedding extraction choices.

2.	Issues regarding the potential overlapping between LM and VM.

3.	How the empirical results on enriched paraphrase reconcile with the large body of prior work stating the human language network does not represent extralinguistic knowledge.

Concerns still outstanding.

1.	The primary unaddressed concern is novelty and framing. Though the authors clarify in the manuscript and the responses, the paper does have a plausible novelty claim; it is narrower than “we discovered abstractness”.

2.	Regarding readability and presentation, the paper should be further organized to include more takeaway arguments that make it easier for readers to recognize the contribution. The manuscript needs further revision before acceptance.

3.	The quality control and human validation aspects remain only partially addressed. The authors’ response emphasizes automated filtering metrics, yet human evaluation is still limited.

4.	The limitation that encoding correlations do not prove modality-independence cannot be fully addressed within this methodology.

**Reviewer Scores:**

Reviewer QaNP: stays 10

Reviewer DsH1: stays 2

Reviewer tWEL: 4 to 6

Reviewer Hsss: likely stays 4

---

### Decision · Program_Chairs · 2026-01-26

Reject